

# Exposure of properties to the 2018 Hurricane Florence flooding: an expanding bull's-eye perspective

Marco Tedesco[1], Steven McAlpine[2] and Jeremy R. Porter[3,4]

1) Lamont-Doherty Earth Observatory of the Columbia University
2) FirstStreet Foundation
3) City University of New York - Quantitative Methods in the Social Sciences
4) Columbia University Medical Center - Environmental Health Sciences

*Correspondence to*: M.Tedesco ([mtedesco@ldeo.columbia.edu](mailto:mtedesco@ldeo.columbia.edu))

**Abstract.** Properly quantifying the potential exposure of property to damages associated with storm surges, extreme weather, and hurricanes is fundamental to developing frameworks that can be used to conceive and implement mitigation plans as well as support urban development that accounts for such events. In this study, we aim at quantifying what was the total value and area of properties exposed to the flooding associated with Hurricane Florence. To this aim, we first generate a map of the maximum flood extent from the combination of the extent produced by FEMA and by means of spaceborne radar remote sensing. Such map is, then, used for estimating the value and area of properties exposed to flooding and the distance of such properties from permanent water bodies. Lastly, we study and quantify how the urban development over the past years and decades over the regions flooded by Hurricane Florence might have impacted the exposure of properties and population to present-day storms and floods, to account what colleagues are starting to address as the "expanding bull's-eye effect" in which "targets" of geophysical hazards, such as people and their built environments, are enlarging as populations grow and spread. Our results indicate that the total value of property exposed was $52B, with this value increasing from ~ $10B (2018 USD) from the beginning of the past century because of the expansion of number of properties. We also found that, despite the slowing of property construction in the decade before Florence, much new construction was in proximity to permanent water bodies, increasing exposure to flooding. Ultimately, the results of this paper provide a tool for the understanding of the approaching reckoning that must take place between our continued development in





coastal areas and the flooding of those areas, which is estimated to increase because of projected increasing sea level rise, storm surges and strength of storms.

## 1    Introduction and Rationale

The projected rise in sea level, increased floods and storm surge and associated consequences over the 21[st] century has the potential to do immense economic harm. The economic impact is particularly worrisome in the U.S. due to the fact that much of the most valuable real estate, densest communities, and most productive economic engines are situated disproportionately in coastal regions

(Fu et. al., 2016; NOAA, 2013; Kildow et. al., 2014).  Generally speaking, sea level rise (SLR) is a long and gradual process that has generally not been thought of as a "now problem" but instead something to be prepared for and discussed as a future issue (Butler et. al. 2016). However, recent research has highlighted an ongoing negative economic signal associated with high-probability flooding events and real estate transactions in coastal communities that can be observed with historical data (see McAlpine

and Porter, 2018; Keenan et. al. 2018; and Bernstein et. al. 2019). This suggests that SLR is a "now problem" and it is already producing meaningful negative economic consequences on coastal communities.  Furthermore, there is ample evidence suggesting that we are only seeing the first signs of a much more problematic issue both in terms of the flooding scale and the magnitude of associated economic losses (see Fu et al, 2016; Hallegatte et al., 2011; Bin et al., 2011; Bin et al. 2008; Parsons

and Powell 2008; Michael 2007). Economic and financial experts often look at damaged areas over large regions, hence missing the details that are necessary to capture the impact of disasters on single unit houses or small areas (see Yohe et al., 1995; Darwin and Tol, 2001; Yohe et al., 1996; Yohe et al., 1999). For example, a sea level rise of six feet would flood roughly 100,000 homes only in New York City, with a total value of $39 billion; a ten-foot rise would flood 300,000 homes and property with a

value of almost $100 billion (UCSUSA, Accessed June 29, 2019). The equivalent figures for Miami are 54,000 homes and property valued at $14 billion at risk with a six-foot rise and 130,000 homes and property valued at $32 billion for a ten-foot rise.



Florence was one of the most devastating hurricanes in history as it combined storm surge, strong winds and extreme precipitation. It began as a tropical storm on 1 September, 2018 over the
Cabo Verde Islands off the coast of West Africa and peaked as a Category 4 hurricane with winds up to 140 mph before making landfall as a Category 1 hurricane on 14 September, 2018 over Wrightsville Beach, North Carolina. By 5 p.m. on Friday, 14 September, 2018 Florence was downgraded to a tropical storm and early on Sunday, 16 September, it diminished to a tropical depression, with winds of about 35 mph.  At least 51 people died as a consequence of flooding associated with rain records (up to
3 feet of rain in some areas according to the Weather Service), leaving more than 400,000 houses without power and generating more than $24 billion in total damage (https://www.ncdc.noaa.gov/billions/events.pdf). The human cost of Hurricane Florence was a reminder of the power of such storms and these storms are likely becoming more impactful as their surge reaches further inland due to changing tracks, increased strength, and rising seas. The response of local
communities to such events can be telling in terms of how the locals are dealing with these trends. Often times an event like Florence has unintended consequences of raising the awareness of the community to all types of flooding.  Such is likely the case in much of the recent research on real-estate market responses to higher-probability flooding associated with nuisance tidal flooding events.

## 1.1 Sea Level Rise and the economics of flooding

The previously cited work by McAlpine and Porter (2018) found that properties in Miami-Dade County at risk of frequent tidal flooding had lost over $430 million in potential property value relative to homes that were not a risk of repeated tidal flooding events.  Likewise, and also centered in the
Miami-Dade region, Keenan et al. (2018) found that homes at lower elevations were being penalized on the market relative to homes at higher elevations.  In a more comprehensive analysis, the research by Bernstein et al. (2018) found a similar penalty for homes at risk of flooding from increases in SLR, but found that this penalty was primarily driven by investors and an uneven access to information associated with risk.  All three of these studies identify an increase in awareness of SLR related
flooding events and all document the fact that this trend is relatively new (since about the middle of the



last decade). Of particular importance to the recent market response is the fact that increased probability seems to be an important driving force. In the work undertaken by Bernstein et al. (2019), the price penalty for homes at risk of flooding is explicitly driven by the sophistication of investors and their access to risk tools aimed at helping them to make decisions about property value, and long-term appreciation over time. McAlpine and Porter (2018) also found that risk associated with being impacted by a Category 1 hurricane is correlated with potential loss property value, but not the probability of being impacted by a higher Category storm. In each of these cases, the research suggests that the real-estate market is becoming more sensitive to the probability of damage associated with inundation from flooding events due to rising seas, storm surges, nuisance flooding and consequences of a changing climate.

Several studies have also recently focused on assessing damages from hurricane Florence. Roberson et al. (2019) use overhead imagery, including synthetic aperture radar (SAR) and optical data, to study the impact of Florence related to livestock wastewaters and to crop health. Srikanto et al. (2019) study the spatial distribution of fatalities and associated demographics, indicating that 93 % of the affected buildings were residential structures. The proper quantification of the impact of Hurricane Florence (or more in general of extreme events) is not only helpful for properly addressing the recovery of the communities impacted by the event but also to provide tools to policy makers, urban planners and city managers that will ultimately guide them through the decision process of reducing the impacts of future events. If it is true, indeed, that climate change is and will be influencing the frequency and strength of storms and floods, it is also true that the impact associated with those events heavily depends on urban development, especially along the coast and in proximity of body waters. Factors such as population growth and the spatial distribution of new properties associated with such growth are key factors for accounting the risks and potential exposure to damage from extreme events.

In this context, it is crucial to study how the urban development over the past years and decades might have impacted the exposure of properties and population to present-day storms and floods. For example, one of the most devastating hurricanes over the Carolinas before Florence was Hurricane Hugo, reaching the Carolinas on 10 September 1989, with winds up to 160 mph and a total estimated damage of $9.45 billion (1989 USD, equivalent to ~ $19B of 2018 USD) and 60 fatalities. Addressing





questions such as: *what would have been the impact of Hugo today or of Florence in the past?* will shed
light on the impact of urban development of properties exposure to such events. Unlike 1989, we have
today improved observational and modeling tools that allow us to better estimate the maximum flood
extent, a key parameter needed to estimate the potential exposure to damage of properties and other
infrastructures. From a modeling point of view, hydrological and hydrodynamic models, in conjunction
with improved digital elevation models and the ingestion of gage observation or observation of high
water marks, offer the opportunity to generate estimates of maximum flood extent (FEMA, 2019).

### 1.2 Purpose of this study

Despite recent studies have started to focus on the spatio-temporal variability of property values
and human settlements in hurricane-prone areas (e.g., Huang et al., 2019) and on the market responses
to increases in observed flooding events (e.g., McAlpine and Porter, 2019; Keenan et. al, 2018), no
study, to our knowledge, has focused on the impact of urban development on the property exposed to
Hurricane Florence. Addressing this point is crucial to start considering not only impacts due to climate
change but also those related to the choices that our society makes to continue the expansion of urban
areas, addressing what some researchers are calling the "bull's-eye expanding effect" (Ashley and
Strader, 2018). Our approach is complementary to those calculating the impact of potential floods under
future, possible climate scenarios (e.g., sea level rise or storm surge is changing but the properties
distribution remains the same). Ultimately, the merging of the knowledge of the spatio-temporal
evolution of properties with future scenarios will allow to identify attributions, allowing to better
estimate damage and risks and supporting urban planning and adaptation strategies. In this study, we
also aim at understanding the usefulness of remotely sensed satellite data as a method for the
identification of impacted areas and for delineating the maximum flood extent. Specifically, we report
results concerning the mapping of the flood extent associated with Hurricane Florence estimated from
SAR data and compare such extent with the maximum flood extent provided by FEMA. From that
exposure, we are able to quantify the property value and total area exposed to Hurricane Florence by
combining the flood extent coverage with a database containing publicly available property value
attributes.



## 2   Data and Methods

### 2.1   Sentinel-1 radar data and identification of inundated areas

From an observational point of view, spaceborne and airborne remote sensing (e.g., Schumann et al., 2011), as well as UAV-based approaches (e.g., Gebrehiwot et al., 2019), offer powerful tools to monitor flood extent (e.g., Domeneghetti et al., 2019; Kordelas et al., 2018; Shumann et al., 2018a, 2018b, Giordan et al., 2018). Optical data can map the presence of surface water at relatively high spatial resolution and accuracy (e.g., Kordelas et al., 2018) but it is limited by the presence of clouds

(Shumann et al., 2018). Datasets collected in the microwave region, such as those collected by Synthetic Aperture Radar (SAR), are not limited by the presence of clouds (Shumann et al., 2018, Manavalan, 2017; Huang et al., 2018). In this case, the recent launch of Sentinel-1 ESA sensors in September 2014 (Sentinel-1A) and April 2016 (Sentinel-1B, https://sentinel.esa.int/web/sentinel/missions/sentinel-1) allows the mapping of flood extent at unprecedented temporal and spatial resolutions. The combination

of the two sensors provides a nominal 6-day repeat cycle over the equator and 12-day repeat cycle over North America (Torres et al., 2012). For the purpose of this study, we downloaded Sentinel-1 data from the National Aeronautics and Space Administration Alaska Satellite Facility (NASA/ASF, https://earthdata.nasa.gov/about/daacs/daac-asf). More information on the Sentinel-1 sensors can be found at https://sentinel.esa.int/web/sentinel/missions/sentinel-1.

Limitations in the case of SAR can arise from features mimicking the behavior of flooded regions (such as some roads or airport runways) and from the presence of natural or manmade structures (such as trees or buildings) that tend to mask the signal associated with the presence of water on the surface (e.g., Shumann et al., 2018; Notti et al., 2018). SAR data has been shown to be able to detect the presence of flooded areas by means of approaches ranging from automated ones (e.g., Bazi et al., 2005;

Moser and Serpico, 2006; Huang et al., 2018; Twele et al., 2016) to sophisticated techniques, such as active contour models (ACM) or change detection (e.g., Landuyt et al., 2019), to simpler ones such as threshold-based approaches (e.g., Shumann  et al., 2018). In this study, we detect flooded areas from SAR assuming that the distribution of the recorded backscattering values can be approximated with a bimodal distribution in which dry (wet) pixels belong to the right (left) normal distribution (e.g., Otsu,



1979; Chini et al., 2017). The threshold value separating the two distributions can be computed through the minimization of a cost function that reflects the amount of overlap between the Gaussian density functions of the foreground and background classes (e.g., Kittler and Illingworth, 1989). As an example, Figures 1a and b shows maps of, respectively, backscattering coefficient values $\sigma_{0VH}$ (Figure 1a, VH polarization) obtained by Sentinel-1A collected on 2 September, 2018 at 23:05:52 UTC and land use

land cover attributes obtained from the National Geospatial Data Asset (NGDA) Land Use Land Cover (LULC) dataset (https://www.sciencebase.gov/catalog/item/581d050ce4b08da350d52363, Figure 1b) from the National Land Cover Database (NCLD, e.g. Jin et al., 2011; Xian et al., 2011) over a subset region where flooded due to Florence occurred. Specifically, we use the "Open Water - All areas of open water" class (Class # 11) to map the presence of permanent water bodies. The original 30 m

dataset is projected on the same coordinate system as the radar images (WGS84) and it is resampled to 10 m through bilinear interpolation to match the SAR data spatial resolution.  Figure 1c shows the histogram (and the fitted bimodal distribution) of the backscattering data in Figure 1a, together with the values of the mean and standard deviation of the two normal distributions, the computed threshold coefficients as well as the Ashman coefficient (See Suppl. Material, Ashman, 1994). This coefficient

allows to automatically quantifying the degree of separation between the two Gaussian distributions: the higher the Ashman coefficient the more the two normal distributions are separated. Here, we adopt the minimum value of 2 on the Ashman coefficient to ensure separability (Chini et al., 2017). For those tiles where the Ashman coefficient is below 2, we use the mean values of the coefficients surrounding that tile. We assess also the use of two ways to compute the threshold value on the radar backscattering

coefficients: the first one ($\sigma_{0Thr1}$) is obtained from the identification of the value where the two fitted Gaussian distributions intersect; the second threshold value is computed as $\sigma_{th2} = \mu_2 - 2*\sigma_2$, with $\mu_2$ and $\sigma_2$ being, respectively, the mean and standard deviations of the "dry" distribution. More details about the SAR-based approach and its assessment are reported in the Supplementary Material.

Once flooded areas are identified, permanent water bodies are excluded using features from the

United States Fish and Wildlife Service's National Wetlands Inventory and the United States Geological Survey's National Hydrography Dataset (https://fwsprimary.wim.usgs.gov/wetlands/apps/wetlands-mapper/). Whenever available, digitized water features from local sources are incorporated in order to



create the most detailed water boundary delineations possible. To produce the final file used to mask permanent water, the classification fields representing deep existing water ("Estuarine and Marine

Deepwater"), Ponds ("Freshwater Ponds"), Lakes, and Rivers ("Riverine") were merged into a single mask. The "Estuarine and Marine Wetland", "Freshwater Emergent Wetland", and "Freshwater Forested/Shrub Wetland" were not included in the original mask as these represent land classification types that are not permanently wet.

**2.2    FEMA Maximum water extent during Florence**

We supplement the radar-derived flood extent with the FEMA's High Water Mark-based Depth Grids and Inundation Polygons from observed and collected Hurricane Florence data. High Water Marks (HWM) are point data collected using high resolution Real Time Kinematic (RTK) GPS systems

or other methods. HWM points represent the highest extent of riverine flood or coastal storm surge inundation. The raw data is available at the FEMA Natural Hazard Risk Assessment Program (NHRAP) site and were downloaded for all basins available per FEMAs collection efforts following the hurricane event (https://data.femadata.com/FIMA/NHRAP/Florence/).

The FEMA Maximum Water Extent is distributed as a GIS raster file created to represent the

extent of riverine or coastal storm inundation following larger flooding events. The file is created as a derived product following the creation of the Maximum Depth Grids raster file, which is created using FEMA HWM data and FEMA's Digital Flood Insurance Rate Map (DRIRM) Base Flood Elevations (LIDAR based elevation data). Using those datasets, a simple grid is obtained from interpolation to estimate the height of water at any given point between HWM based on base elevation. From this, we

extracted a secondary file measuring only the extent of inundation from the storm surge. The FEMA dataset is distributed as an ARCGIS® geodatabase (.gdb format) and it is geolocated and rasterized at a spatial resolution of 10 m to match the spatial resolution of the SAR data. More information on the FEMA    approach    for    estimating    maximum    flood    extent    can    be    found    at https://data.femadata.com/FIMA/NHRAP/Florence.





### 2.3 Property database

Property value data is compiled from each individual property's county assessor in the form of the property tax assessed value. The data were obtained from a third party provider, ATTOM[TM] Data Solutions, which provides high quality parcel level information on all properties in the United States

and in a value added format (https://www.attomdata.com). The process by which the data are compiled relies solely on publicly available data and the processing, cleansing, standardizing of that data in order to make it available in a user-friendly format. The data used in this analysis include the property's last recorded assessment value for all properties within the states of North and South Carolina. Each county's assessment process varies and, as such, the data are subject to known potential limitations

associated with the timing and frequency of home assessments undertaken by local county officials in which the property is located. However, the data also give us the best available comprehensive look at tax base value in a geo-located format for comparison to our storm surge coverage file. Beside property values, the database also contains the year when each property was built, which we use for our expanding bull's-eye effect analysis.

## 3  Results and discussion

### 3.1  Assessment of remote-sensing derived areas vs. FEMA maximum water extent

Inundated areas (including permanent water bodies) obtained from Sentinel-1 data are reported as blue regions in Figure 2a, together with the maximum water extent estimated by FEMA (red areas). We used a total of 12 Sentinel-1 images collected between 14 September and 19 September, 2018 and

whose footprints are shown in the inset in the top left corner of Figure 2b. Specific names and acquisition times of the radar images are reported in the Supplementary material for reader's convenience. We used the 12 images in order to maximize the covered area and to account for the temporal evolution of surface water after the landfall of Hurricane Florence associated with heavy, persisting rainfall.

The comparison between the maximum water extent estimated by FEMA and the water extent mask obtained from Sentinel-1 indicates a matching score (defined here as the percentage of flooded





pixels identified by Sentinel-1 with respect to the total number of flooded pixels identified by FEMA) of 11.3 % and a commission error (defined as the relative percentage number of pixels when Sentinel-1 detects flooded areas but FEMA does not with respect to the total number of FEMA flooded pixels) of

9.2 %. When considering the two maps obtained from the two approaches, we have to remember that FEMA water extent map is based on a combination of modeled and measured quantities, as explained above and it is, therefore, possible that some areas that were included in the radar images were not included in the FEMA maps as they occurred away from the ocean water. As an example, Figure 3 shows the maximum water extent from FEMA (red) together with the water extent derived from

Sentinel-1 data nearby the town of Bennetsville, NC (34.6174° N, 79.6848° W). Green dots show the distribution of properties from our database. If on one side Sentinel-1 is missing to capture the maximum water extent estimated by FEMA, it is also true that the radar sensor is suggesting the presence of flooded areas (mostly agricultural fields) that are not included within the FEMA extent. Our analysis of the Sentinel-1 backscattering coefficients (not shown here) indicates that the backscattering

values recorded for those regions where flood was identified were relatively low (e.g., well below the threshold value and on the order of ~ -20 dB or below), strongly suggesting that those were inundated areas. We argue that for such areas the FEMA approach might not have captured those flooded areas, either because of limitations related to the hydrodynamic model used in the approach or because of the lack of collection of in-situ data. Another factor complicating the comparison between Sentinel-1 and

FEMA inundated regions is that the acquisition times of the radar images do not coincide with the time of the maximum water extent associated to storm surge. Figure 4a shows the time series of the water height (mean sea level in meters) for the ocean tide gauge located in Wrightsville Beach, NC (id #8658163), where Hurricane Florence made landfall. Maximum water height was reached on the same day around 15:00 UTC. The image also shows the acquisition times of the Sentinel-1B (14 September,

2018, 11:15:05, UTC) and Sentinel-1A (14 September, 2018, 23:05:48, UTC) as vertical, dashed lines, indicating that the images were, unfortunately, acquired before and after the maximum water height. On the other hand, river gages data show that the maximum water discharge and gage heights inland occurred a few days after hurricane Florence made landfall, because of the heavy precipitation. In this regard, Figures 4b and 4c show, respectively, the daily discharge (in cubic feet per second) and daily





gage height (in feet) recorded at the river gage station of Lumberton, NC (34.6182° N, 79.0086° W), located about 150 km inland. The data shows the peak discharge and water heights late in the evening of 17 September, 2018. For this area the radar data were collected when the tide gage recorded peak values, confirming the usefulness of this tool to capture flooding that FEMA might have been missing that. As a further example, we show in Figure 5 the flooded areas detected by Sentinel-1 (blue filled

regions) on 19 September, 2018 nearby Pasley, Duplin County, NC (34.7854° N, 77.9005° W) and a photograph of the same area collected on 18 September, 2018 by the NOAA Remote Sensing Division to support emergency response requirements (https://storms.ngs.noaa.gov/storms/florence/index.html#7/35.360/-77.820). Most of the flooded areas identified by the NOAA photograph are properly captured by Sentinel-1, with differences between the

two also due to the different acquisition times. Nevertheless, for this area, the FEMA map does not indicate any flooding, confirming the complementary nature of the radar dataset.

Given the above considerations, we merged the FEMA and Sentinel-1 flood extent maps to generate a maximum composite flood extent map that will be used to assess the property exposure to Hurricane Florence flooding. We will refer to this dataset simply as the "maximum flood extent" in the

remaining sections of the manuscript.

### 3.2 Exposure of property to Hurricane Florence flooding

Figure 6 shows the spatial distribution of the properties within our database overlaid with an image of the eye of Hurricane Florence when it made landfall. Our results indicate that the total area of

properties affected by the maximum flood extent water was 70,964,700 m$^2$, being 17.55 % of the total area within our database. When considering only the flood extent estimated by Sentinel-1, the total area of properties affected by the estimated flood extent reduces to 3.2 %, corresponding to 12,939,432 m$^2$. In order, to quantify potential biases associated with co-registration issues or resampling procedures, we computed the number of properties exposed to the extent of our permanent body water dataset. Our

analysis shows that less than 0.2 % of properties was overlapping with the permanent body waters. Consequently, we removed these properties from our analysis.



We estimate a total property value exposure of \$52,079,520,584 (corresponding to ~ 9.5 % of the total property value within our database), of which \$9,437,931,512 when considering only Sentinel-1. The exposed property value identified by Sentinel-1 but not by FEMA was \$3,278,098,601. Beside

the limitations discussed above, the relatively small exposure area and property values obtained with Sentinel-1 are also due to missing detection of flooded regions in urban areas by radar (e.g., Notti et al., 2018). Indeed, the basic assumption for detecting flooded areas by Sentinel-1 is violated by the presence of dense vegetation or buildings. The radar signal will, in this case, bounce on the vertical structures (e.g., buildings and trees) after being reflected by the water surface, increasing the amount of energy

reaching the radar receivers instead of reducing it and, therefore, increasing the value of the backscattering coefficient. In these cases, therefore, despite the presence of flooding on the surface, the backscattering values will not belong to the "wet" Gaussian distribution and the masking effect of buildings and trees will misclassify those areas as dry (e.g., Schumann, 2018a, 2018b). Another reason for the underestimation of property exposure derived from Sentinel-1 data can be seen in Figure 3,

where it appears evident that Sentinel-1 is detecting flooding over rural and agricultural areas (and FEMA is not) where the number of properties is relatively smaller than highly density populated areas.

In Figure 7 we report the distribution of the number of properties exposed to flooding within our database as a function of property value. A power law function as reported in Eq. 1

$$Y = a*x^n \tag{1}$$

fitting the histogram is also plotted as a dashed, black line with *a* and *n* obtained from the fitting as $a = 1.9544*10^6$ and n = -1.1216. The power law function here selected was chosen after testing several functions (e.g., exponential decay, logarithmic, etc.) as the one showing the highest regression

coefficient (*R = 0.99*). We remind to readers that the property value used to compute the coefficients is expressed in thousands of dollars. According to Zillow©, the median home values in North Carolina and South Carolina are, respectively, \$184,200 (North Carolina) and \$166,300 (South Carolina) with a median price of homes of \$196,600 in the case of North Carolina (https://www.zillow.com/nc/home-values/) and \$178,800 for South Carolina (https://www.zillow.com/sc/home-values/). We use these



estimates to set to $200k the median price within our database and evaluate the number of properties this value using Eq. 1, finding that 40 % of the properties exposed to Hurricane Florence flooding were below the selected value. The properties valued between $200k and $500k account for another 25 % whereas the properties with values between $500k and $1M account for another 25 %. As a reference, the total number of properties valued below $200k represent ~ 50 % of our database, those between $200k and $500k are ~

25 % and those between $500k and $1M roughly 15 %.

    Distance from water bodies, especially coastal and riverine bodies, is also a useful indicator of properties vulnerability and potential exposure in hurricane prone areas. Therefore, we expanded our analysis to consider the distance of the properties that were flooded during Florence within our database from permanent water bodies (Figure 8). Values along the x-axis in the plot are obtained as the

minimum distance from any of the closest element of the permanent water bodies mask (e.g. ocean, rivers, lakes) to each property in our database. The figure also shows the exponential decay function fitting the histogram and the fitting parameters. From this, we estimate that ~ 95 % of the number of properties exposed to flooding fell within 10 km from body waters. This number increases when considering only the distance from the ocean because of the inland flooding associated with heavy

precipitation. We, therefore, use the distance of 10km as a maximum distance to be considered for studying the relationship between new properties, their distance from water bodies and the exposure to the Florence flood extent.

### 3.3   Impact of expansion of urban areas on property exposure

    As mentioned in the Introduction, the exposure to floods and other extreme events depends not

only on the geophysical hazard but also on how urban growth and infrastructures have been, are and will be evolving in the areas at risk. This concept is well synthesized in what has been named "the expanding bull's-eye effect" (Ashley and Strader, 2016), arguing that ''targets''—people and their built environments— of geophysical hazards are enlarging as populations grow and spread. In order, to investigate the impact of the expanding bull's-eye effect on the property exposed to the flooding of

Hurricane Florence, we calculated what would have been the property area and values exposed to the Florence flood should that have occurred 10, 50 or 100 years ago by using the information contained




within our database on the years when properties were built. For the purpose of this analysis, we clarify that we are assuming the same sea levels and topography of today.

Figure 9 shows the spatial distribution of the properties within our database that were built during the a) 1800 – 1900, b) 1900 – 1950, c) 1950 – 2000 and d) 2000 – 2018 periods. We considered the first period as a 100-year one (Figure 9a) because of the relatively small number of properties that were built then. Most of the development between 1900 and 1950 (Figure 9b) occurred inland and along the coast north of Wilmington, with a relatively small number of new properties built close to water bodies (either rivers or ocean). An explosion in new properties occurred between 1950 and 2000 (Figure 9c), likely as a consequence of the economic stimulus following World War II. The period 2000 – 2018 shows a relatively smaller number of new properties with respect to the previous periods (Figure 9d). This is partially related to the shorter period considered for the last panel. However, our analysis performed on the 10-year period of number of properties built within our database (Figure 10) shows that before 2010 the number of houses built had been increasing exponentially ($Y = 5e\text{-}22 * \exp^{0.0314*X}$, R = 0.99, with X being the year) and that the number of new properties after 2010 drastically dropped, reaching values similar to those observed before the 1950s. This might be due to the 2008 "house crisis" that occurred during that period.

Figure 11 shows the time series of total value of exposed property (in 2018 $B). The inset shows the relative change of the exposed area and value between two consecutive time steps (10 years). Consistent with the results discussed above, a relatively small increase in the exposed property value occurs before the 1940s (from ~ $10B to ~ $12B). The increase becomes substantial after 1940s, driven by the building of new properties (Figure 9), reaching a maximum value of exposed property of ~ $52B in 2018. We fitted the increase in exposed property value after 1900 with an exponential function ($Y = a*\exp^{bX}$) and computed the coefficients providing best fitting ($a = 1.0627*1e^{-13}$, $b = 0.167$, R = 0.97). The maximum relative increase is reached around the year 2000 with a relative increased exposed value of ~ $8B between two successive decades. After then, the relative change in exposed property values decreases to values close to those obtained in the early 1950s.

As mentioned above, distance from permanent water bodies can play a critical role in terms of exposure, with flooding due to Hurricane Florence reaching properties that were up to ~ 10 km from the



90  closest water body. Therefore, we further studied how the property value evolved as a function of the
distance from water bodies between 1800 and 2018. As an example, Figure 12 displays the distribution
of properties built during different periods in proximity of Wrightsville beach, where hurricane Florence
made landfall, highlighting the expansion of urban areas along the coasts and water bodies especially
between 2000 and 2018. Figure 13 shows the total value of exposed properties within our database as a

95  function of distance from water bodies between 1800 and 2000 (using a 25-year time step) and for the
period 2000 – 2018. As it is possible to observe from the figure, the curves referring to early periods
reach a plateau within a relatively short distance than those referring to later periods, with the saturation
values (e.g., the value when the curve becomes flat) being of the order of 1500m in the case of the 1975
– 2000 period. Interestingly, the period 2000 – 2018 does not show a plateau but the exposed property

values continue to increase as the distance from water bodies increase. Therefore, despite the most
recent decades were characterized by a relatively smaller number of new properties (Figure 10), the
potential exposure to Florence of such properties was higher because of the higher number of the
exposed properties close to body waters.

## 4    Conclusions


Increased flooding associated with sea level rise, storm surges and other extreme events has the
potential to disrupt economically many areas around the world, with most of valuable real-estate,
densest communities and most productive economic engines situated in coastal regions. The specific
goal of our study was to quantify the exposure of properties to the flooding associated with Hurricane

Florence that hit the Carolinas in September 2018 and to study how the spatio-temporal evolution of
new properties during the past century and most recent decades has impacted the property exposure.
Indeed, if it is true that risk and exposure to events arise from the geophysical processes (e.g.,
hurricanes, rainfall, etc., e.g., Stone and Cohen, 2017), it is also true that the choices associated to where
to build new properties can have a profound effect on the impacts and risks through the so-called

"expanding bull-eye's effect". Despite these considerations appear to be obvious, very few studies have
focused on this aspect and, to our knowledge, this is the first study focusing on this for Hurricane




Florence. In order, to properly quantify the exposure of properties to Florence flooding, we developed a maximum flood extent map from the combination of the FEMA maximum extent map generated through the merging of high water marks and the outputs of a model and the flooded areas detected by

means of spaceborne radar data acquired by the ESA Sentinel-1 sensors. We found that Sentinel-1 data underestimates the maximum flood extent with respect to FEMA because of both intrinsic (e.g., loss of sensitivity to flood over dense urban areas) and extrinsic reasons (e.g., acquisition time not overlapping with the timing of maximum flood extent). However, we also found that radar data can detect flooding over rural and agricultural areas that is not detected by FEMA, hence filling the potential gaps of the

FEMA approach. We, therefore, combined the two maps to obtain an optimal maximum flood extent map that we used for our quantitative analysis.

We found that the total value of property exposed to flooding was ~ \$52B and that this value has increased exponentially from ~ \$10B (2018 \$US) in the early 1900s. This is due to the increase in the number of properties that came to a halt at the beginning of the 2000s, likely as a consequence of the

2008 housing crisis, when the number of new properties built after 2010 was almost half of those built only a decade before. Despite this, the exposure to Florence flooding for those properties built after 2000 continued increasing, because of the number of new properties built within proximity of permanent water bodies and coastlines.

Our work cannot only provide new insights for policy makers and city planners but it also does

provide a tool to better estimate how the property market will respond to future disasters. Recent work (e.g., McAlpine and Porter, 2018; Keenan et al., 2018) has found that homes at lower elevations were being penalized on the market relative to homes at higher elevations and that houses exposed to sea level rise (SLR) sell for approximately 7% less than observably equivalent unexposed properties equidistant from the beach (Bernstein et al., 2019). For our future work, we plan to expand our analysis

to other modern-day (e.g. Irma, Michael, Katrina and Sandy) and historical (e.g. Hugo in 1989) hurricanes to address similar questions to those addressed in this study. Moreover, we plan to improve the detection of maximum flood extent through the implementation of machine-learning techniques combining radar maps with tide gage interpolated data and other ancillary information. Lastly, the combination of the knowledge on how property distribution changed along the years in conjunction





with outputs of physical or probabilistic models that can separate the different contributions associated to flood due to SLR, storm surge and rain will allow to properly quantify what is the impact of the different components of the climate-economic system on the total exposure and, eventually, damage. This will provide a crucial tool for policy makers, governments, citizens and those who are, rightly, interested in quantifying the impact of climate change on the economic and house markets.


**Authors contribution.**

MT conceived the study and wrote the first draft of the manuscript. JP and SM provided and analyzed the property data, permanent water bodies and LULC data. MT developed and implemented the code
for the radar dataset. All authors contributed to the final analysis and final version of the manuscript.

**Acknowledgments.**
MT acknowledges financial support from Columbia University through the RISE program and from FirstStreet foundation.


**Code availability.** We used a combination of publicly available software and codes developed ad-hoc for the purposes of this study. Specifically, the ESA SNAP software used to pre-process the Sentinel datasets is available at http://step.esa.int/main/download/snap-download/. We also used QGIS 3.4 to export the property data into a shapefile and to analyze the permanent body waters and the FEMA
maximum flood extent data. The software is available at https://qgis.org/en/site/forusers/download.html. We developed in-house codes in Matlab for mapping flooded areas from radar data and to perform the analysis of the exposed property values. These are available upon request to the corresponding author at mtedesco@ldeo.columbia.edu

**Data availability.** Sentinel1 data is freely available at https://earthdata.nasa.gov/about/daacs/daac-asf. The dataset containing the permanent water bodies is available at https://fwsprimary.wim.usgs.gov/wetlands/apps/wetlands-mapper/. Land use land cover attributes



obtained from the National Geospatial Data Asset (NGDA) Land Use Land Cover (LULC) dataset is available at (https://www.sciencebase.gov/catalog/item/581d050ce4b08da350d52363. Maximum water

extent by FEMA is available at https://data.femadata.com/FIMA/NHRAP/Florence/. Property value data is compiled from each individual property's county assessor in the form of the property tax assessed value and was obtained from ATTOM™ Data Solutions. Those interested in this dataset should reach out to the corresponding author mtedesco@ldeo.columbia.edu or can be obtained at www.attom.com.


**Competing interest**. The authors declare no competing interest.



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

00

05

## 5    Figures

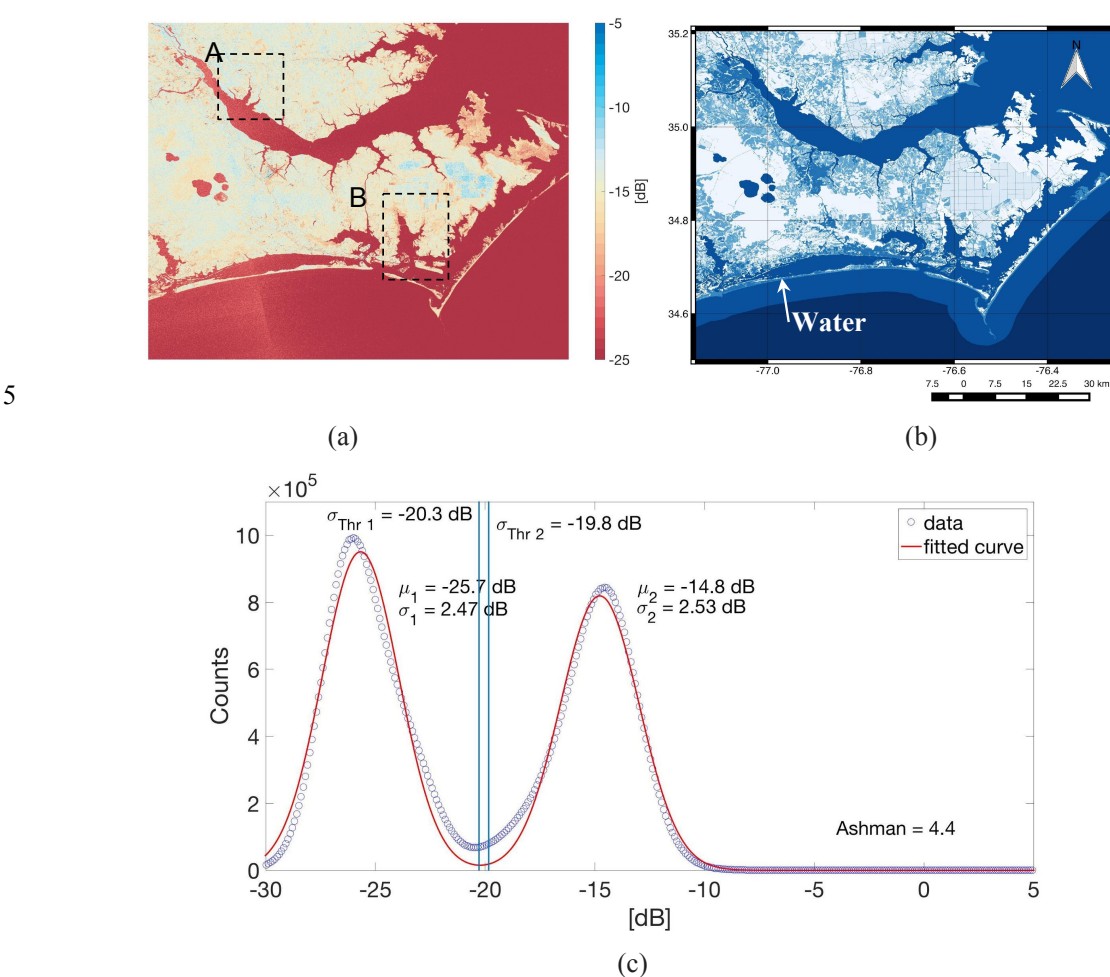

(a)                    (b)

(c)

**Figure 1** Map of VH backscattering coefficients obtained from Sentinel-1 collected on 2 September, 2018 at 23:05:52 UTC over a
20    region impacted by Florence b). Same as a) but showing land cover attributes from the National Geospatial Data Asset (NGDA)
Land Use Land Cover (LULC) dataset data record (https://www.sciencebase.gov/catalog/item/581d050ce4b08da350d52363). Here,
water is represented as medium blue. c) Histogram and fitted bimodal distribution of the backscattering data in a). In the panel,
the mean and standard deviation of the two normal composing the bimodal distributions are reported, together with computed
thresholds and the Ashman coefficient.

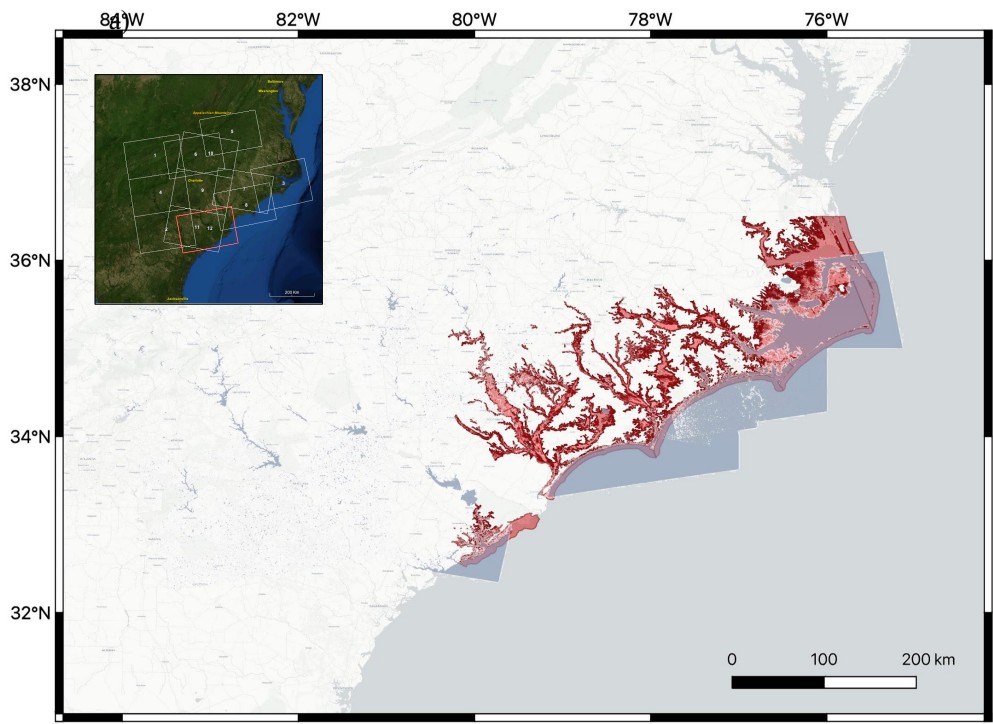

**Figure 2 Map of inundated areas estimated by FEMA (red) and by the Sentinel-1 radar images (blue). The inset in the top left corner shows the footprint of the several radar images to create the composite water extent map. Acquisition times and other details concerning the radar images are available in Supplementary material.**

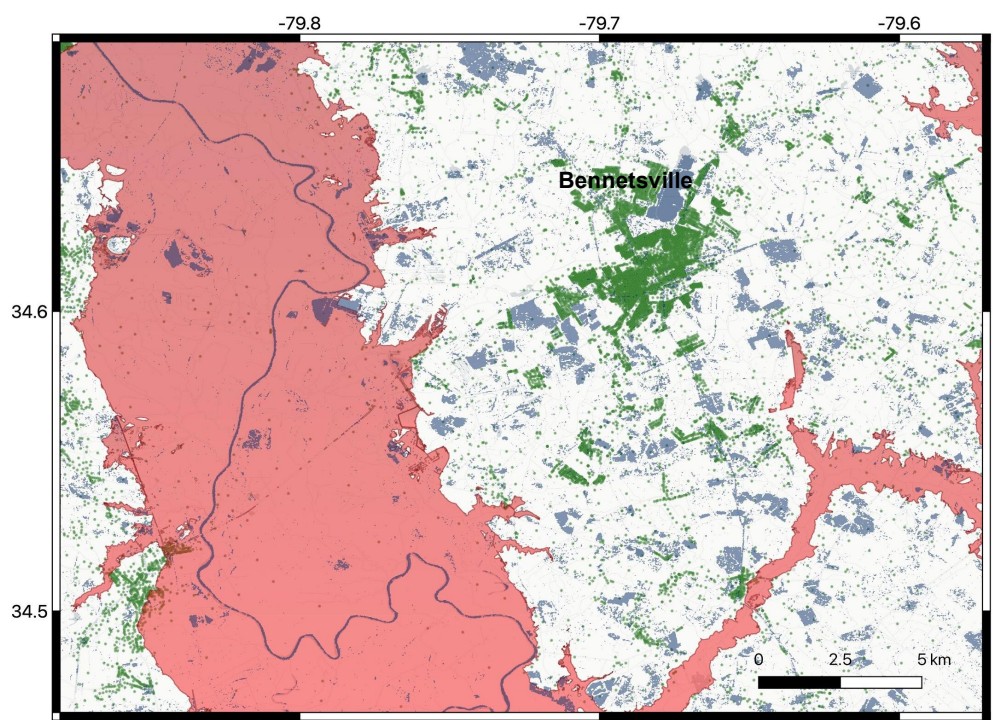

**Figure 3 Map of inundated areas estimated by FEMA (red) and by Sentinel-1 (blue) near the town of Bennetsville, SC (34.6174° N, 79.6848° W). Green dots represent the locations of properties for this area.**


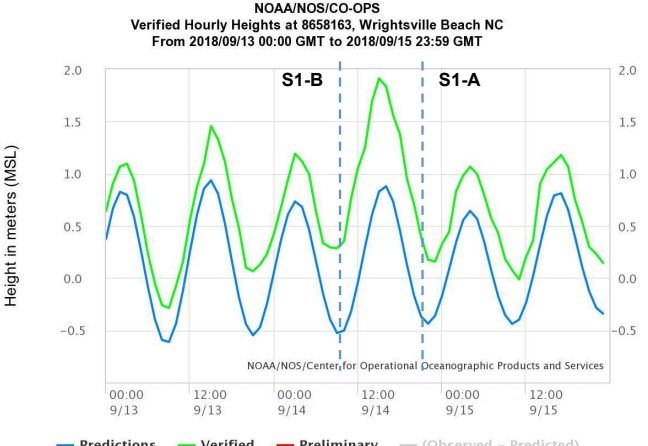

(a)

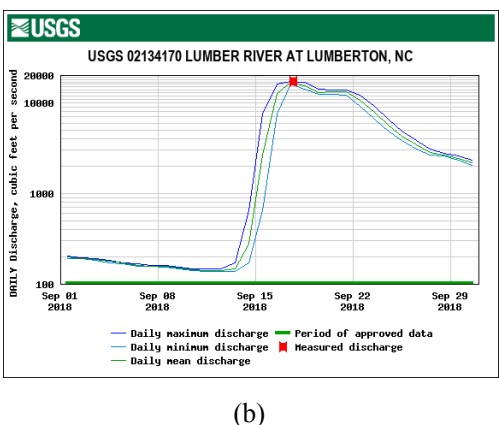

40                          (b)

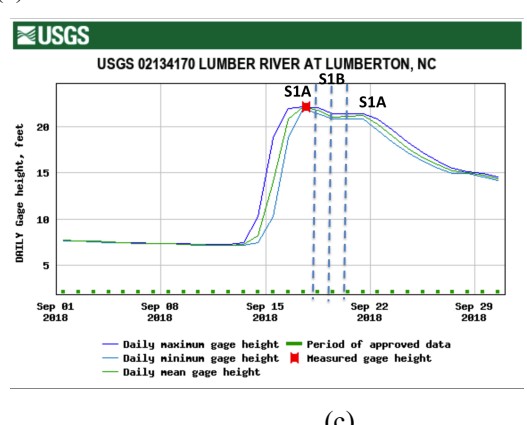

(c)

**Figure 4 Time series of a) tide gage mean sea level height (meters) recorded at Wrightsville Beach, NC and b) daily discharge (cubic feet per second) and c) daily gage height (feet) recorded at Lumber river (USGS gauge 02134170), NC between 1 September and 30 September 2018. In a) blue line refers to predictions where green squares to verified values. In a) data and plot was obtained from https://tidesandcurrents.noaa.gov/. For data plotted in b) and c) we obtained data and graphs from https://waterdata.usgs.gov/. In a) and c) we also report as dashed vertical lines the acquisition times of the available Sentinel-1 data.**



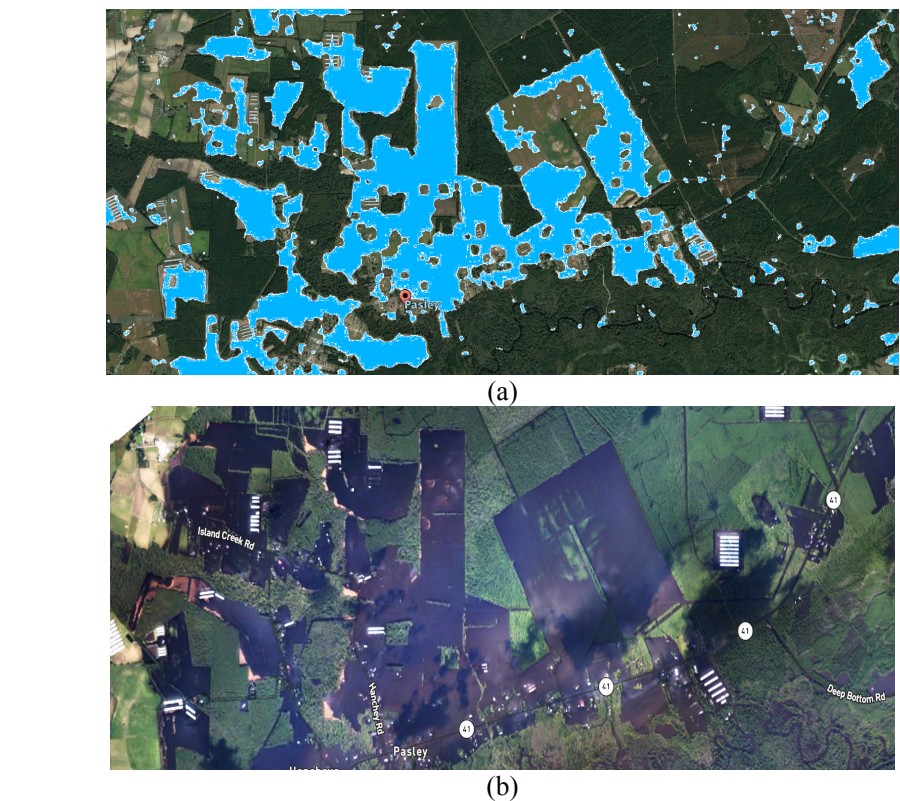

(a)


(b)

**Figure 5 Flooded areas detected by a) Sentinel-1 data (light blue filled regions) on 19 September, 2018 nearby Pasley, Duplin County, NC (34.7854° N, 77.9005° W) and b) photograph of the same area collected on 18 September, 2018 by**
**NOAA (https://storms.ngs.noaa.gov/storms/florence/index.html#7/35.360/-77.820). Here, dark blue regions show flooded areas.**
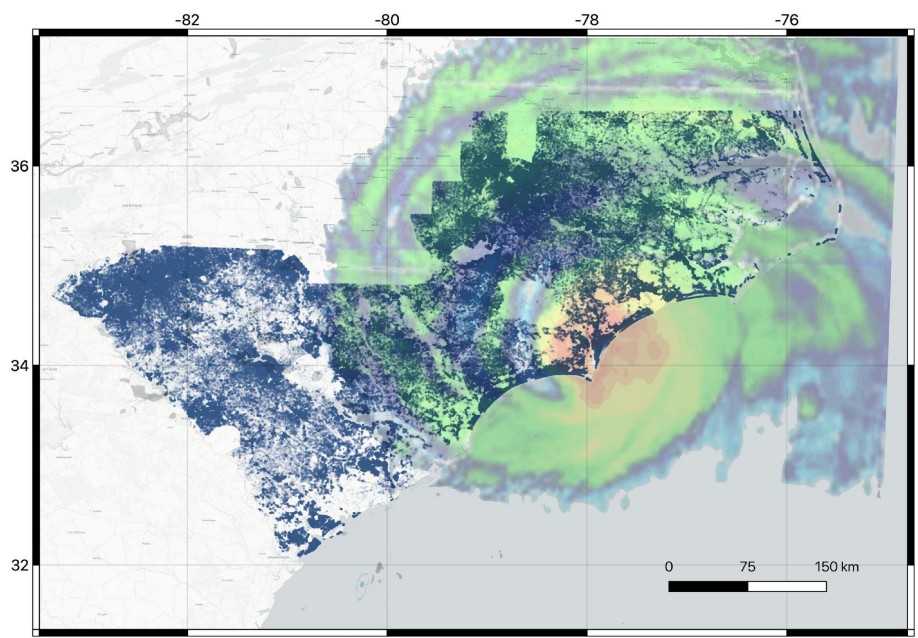

**Figure 6 Distribution of properties within our database used to estimate the exposed property damage to Florence Hurricane. An image of the Hurricane Florence making landfall is also reported as a reference (Hurricane image courtesy: Cyclocane).**


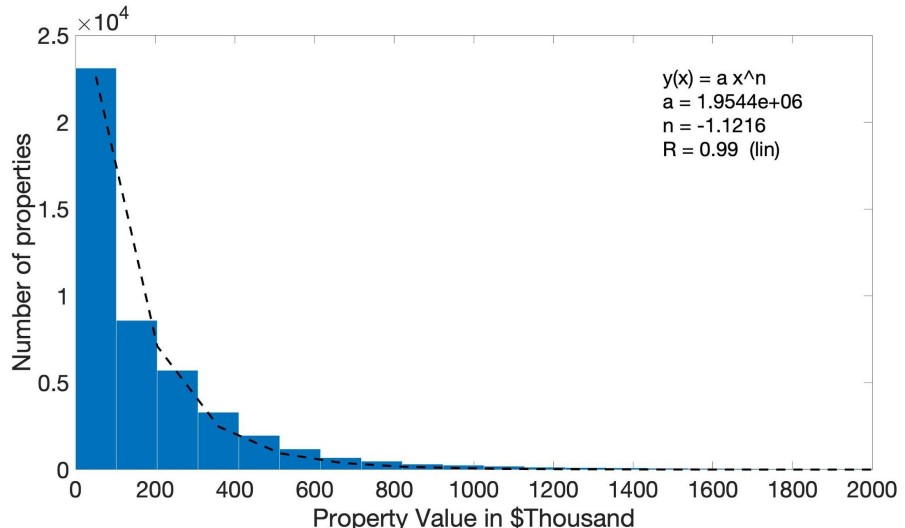

**Figure 7 Distribution of the number of properties exposed to flooding as a function of property value. Dashed line represents the power law curve fitting the distribution. The parameters of the fitting power law function are reported in the top right section of the figure.**


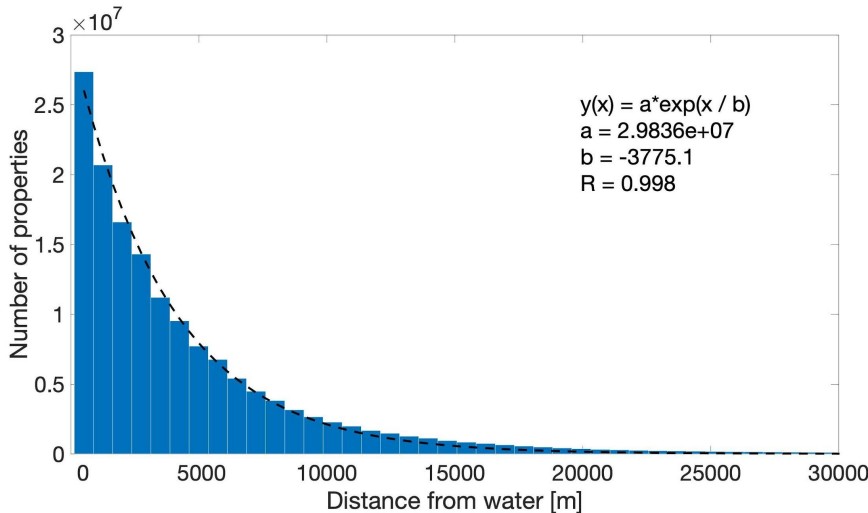

**Figure 8 Number of properties as a function of distance from water bodies. Dashed line represents the power law curve fitting the distribution. The parameters of the fitting power law function are also reported in the top right section of the figure.**




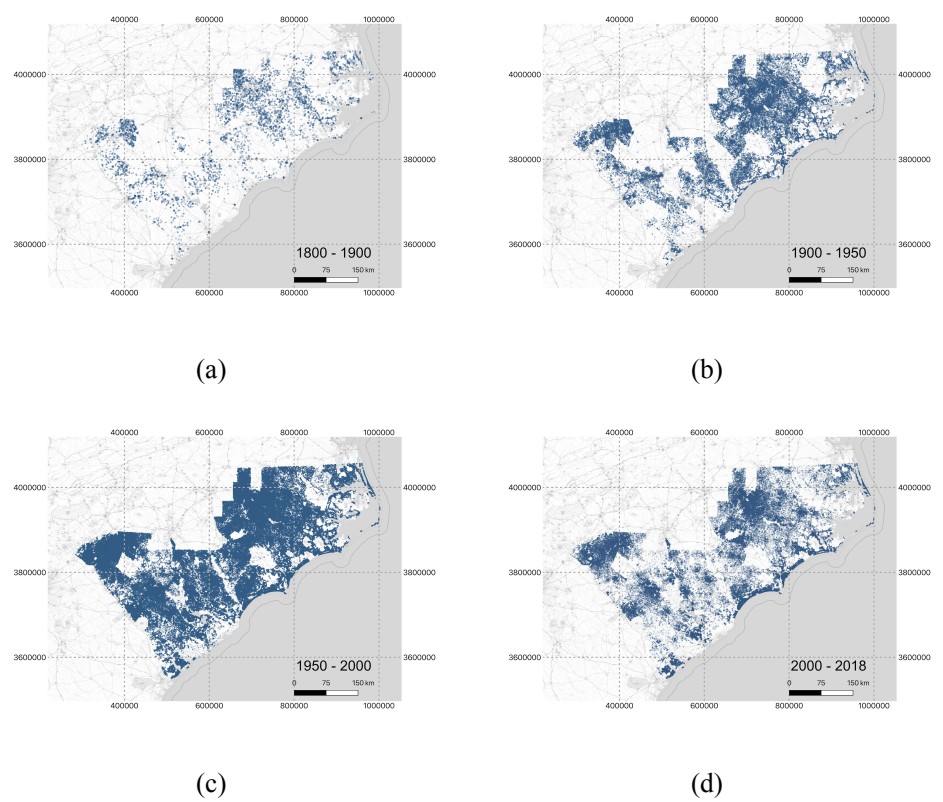

(a)  (b)

(c)  (d)


**Figure 9 Spatial distribution of the properties within our database that were built during the a) 1800 – 1900, b) 1900 – 1950 , c) 1950 – 2000 and d) 2000 – 2018 periods.**
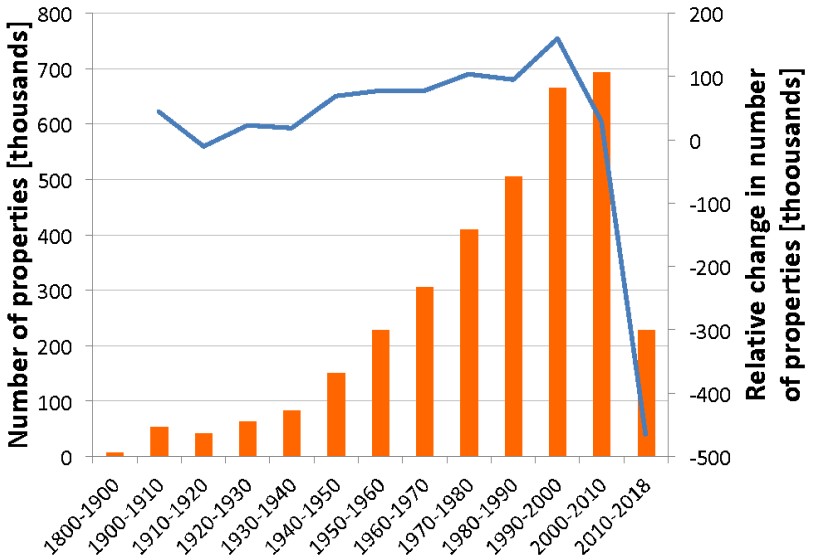


**Figure 10 Number of properties (in thousands) built within our data record during different decades (red bars, left axis) and relative change between two consecutive periods (blue line, right axis). Note that the number of properties built between 1800 and 1900 are aggregated as a single value because of the small number of properties built during that period.**




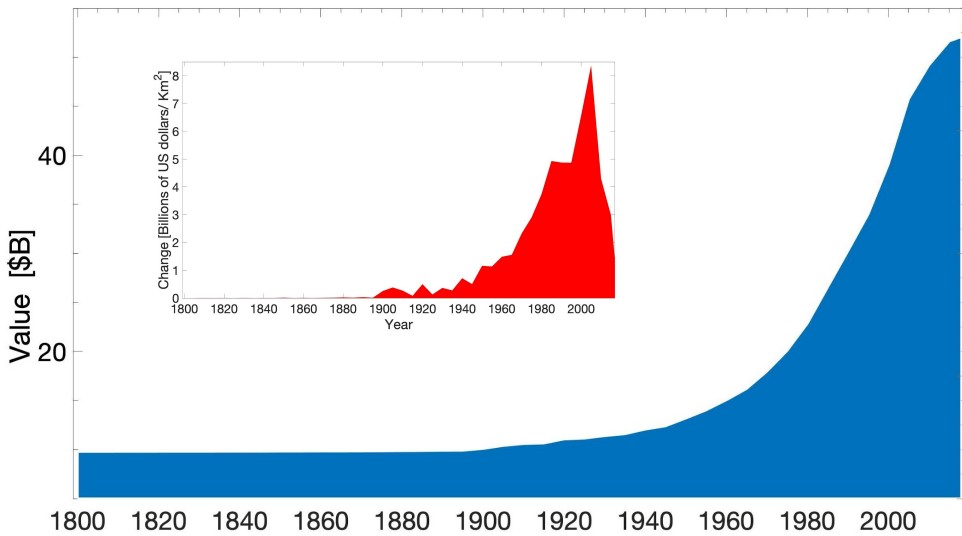

**Figure 11 Time series of total value of exposed buildings (in $B) to the maximum flooded extent region between 1800 and 2018. The inset shows the relative change of the exposed area and value between two consecutive time steps (10 years).**


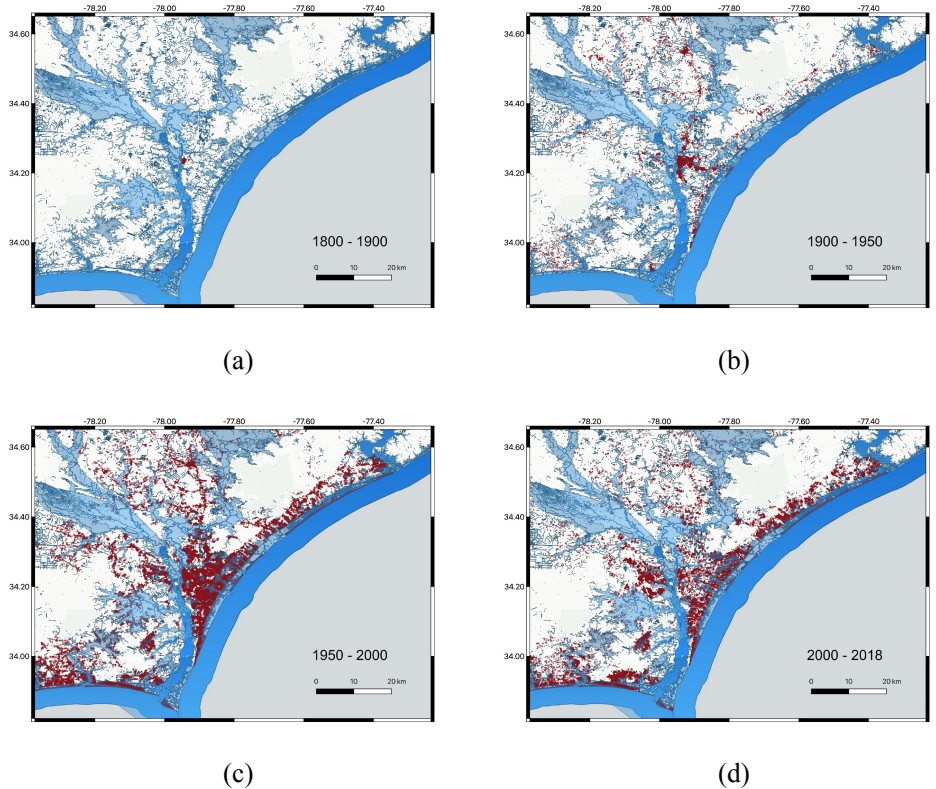

|  |  |
|---|---|
| (a) | (b) |
| (c) | (d) |

**Figure 12 Distribution of properties (red dots) built a) before 1900, b) between 1900 and 1950, c) between 1950 and 200 and d) between 2000 and 2018 in proximity of Wrightsville Beach, NC where Hurricane Florence made landfall. Dark blue shows permanent body waters where light blue shows the flooded areas.**

00

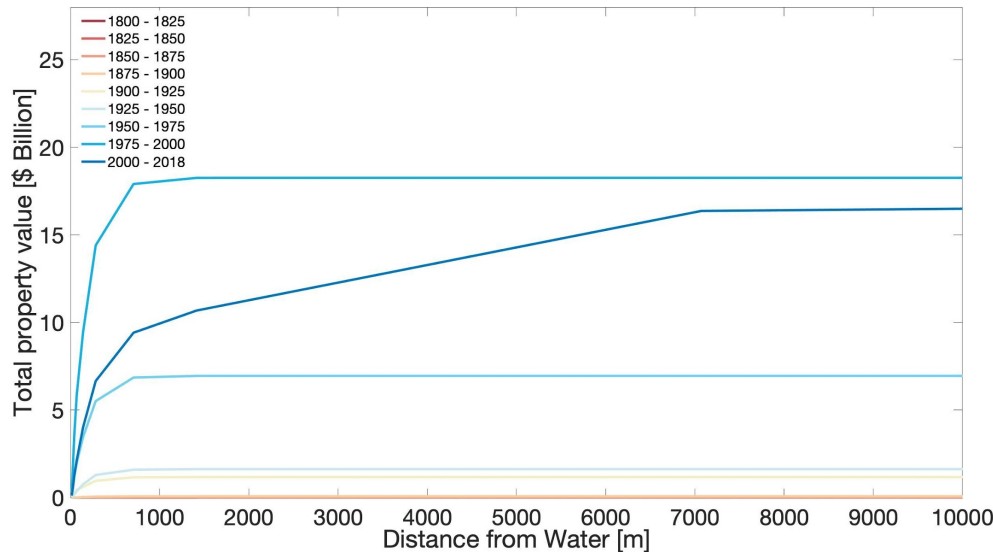

**Figure 13 Total value within our database of properties exposed to flooding as a function of distance from water for the different periods reported in the inset.**

05