# Peer review of "Exposure of real estate properties to the 2018 Hurricane Florence flooding"

_Natural Hazards and Earth System Sciences, 2019_

## Referee Comment (RC1) · Anonymous Referee #1 · 30 Jul 2019

Please see the following commentary:

1. General: The writing style should be more formal and should rely more on active verbs. In general, the manuscript should be edited down to simplify the sentence structure. In addition, there are many errors of punctuation and grammar that are distracting.

2. Findings: The inadequacy of FEMA flood mapping should be reinforced and contextualized. For instance, how can this method augment or support alternative determinations of spatial risk in the housing market? I would like to see some analysis of the method and findings within matters of local and federal policy. For instance, could this

method compliment Hazus? Maybe or maybe not. At its core, this is about land use and zoning and you should be explicit about this.

3. Line 45: There is also some counter- and supporting-evidence on the horizon that is worth reading:

Gibson, M., J. Mullins, and A. Hill 2017. "Climate Change, Flood Risk, and Property Values: Evidence from New York City". Working paper, University of Massachusetts-Amherst.

Murfin, J. and M. Spiegel 2018. "Is the Risk of Sea Level Capitalized in Residential Real Estate?" Review of Financial Studies Climate Finance Call, Conditionally Accepted.

4. Present Value: When citing valuations, it is always good practice to cite the associated dollar year.

5. Lines 58-70: This should be significantly edited-down. I think it is also useful to highlight the economic distinctions between stresses and shocks because they have different theoretical pricing elasticities.

6. Line 86: "Of particular importance to the recent market response is the fact that increased probability seems to be an important driving force." A "fact" and "seems" are contradictory language here.

7. Lines 103-108: It is critical that you use the precise language of exposure and sensitivity within the conceptual framing of vulnerability. Much of your pivot argument around urban development is fundamentally about exposure that has merit independent of climate attributed phenomena.

8. Dates: We only need years–not days and months (unless you are referencing SAR).

9. Metric System: I would recommend converting to the metric system.

10. Line 130: I would go ahead and define the "bulls-eye."

11. Line 138, et seq.: Please define acronyms in their first usage.

12. Section 2.3: Are there other data sources from which you can triangulate and validate assessed value with market value to normalize across the distribution to account for varying appraisal methods and time limitations? I'm not sure I understand how this relates to the Zillow data that you later cite.

13. Page 14, Line 369-70: "An explosion in new properties occurred between 1950 and 2000 (Figure 370 9c), likely as a consequence of the economic stimulus following World War II." This is not a sound association over the cited length of time. Economic research would suggest that "half-back" population shifts associated with these areas being low-cost retirement areas for high-cost Northeaster residents is the primary driver of coastal development patterns here.

14. Page 5, Line 125: " no study, to our knowledge, has focused on the impact of urban development on the property exposed to Hurricane Florence." I suppose this is true, but the real contribution is the time-space distribution effects on absolute exposure– independent of climate change attributed events. This exposure has happened and exists whether or not climate change impacts are measured or not.

15. Page 14: If you could find a weighted average assessment tax, it would be helpful to contextual how important these properties are to local tax bases (county-level). The raw numbers are hard to contextualize.

16. Page 16: When you cite the increase in valuation, my first question is: so what? You really need to another metric for contextualizing exposure. Citing 1940s valuation is methodologically not particularly sound anyway. Again, I recommend looking at local tax base and operating in percentage share.

17. Overall: This paper needs to be significantly edited and simplified. The work is largely solid, but it meanders too much. The connections to climate change are somewhat disconnected as a matter of attribution. The value is the observation method

and the context of expanding exposure. The science appears to be well-grounded (beyond my competency to full evaluate), but the economics needs some work.

---

## Referee Comment (RC2) · Anonymous Referee #2 · 7 Sep 2019

**REVIEW – nhess-2019-209 (Tedesco *et al.*)**

This manuscript aims to quantify an "expanding bull's-eye effect" in the densely developed region of the US Atlantic Coast hit hardest by Hurricane Florence in 2018.

Overall, the concept for the article is engaging, and represents a useful analysis of how exposure of physical property to coastal flood risk has changed over time.

I have some broad suggestions (organised by line number) for the authors that I hope might improve the manuscript. I do think there is one major issue (see remarks on Section 3.3, below) that will need to be resolved before this work is ready for publication.

- L45 – the "now problem" phrase is cumbersome and seems unnecessary. Consequences of flooding from sea-level rise has been a fixture of IPCC reports, etc – I think the authors should simplify/clarify this rhetorical line or cut it for a straighter delivery.

- L50 – I don't understand the counterpoint in this sentence/paragraph. This work is also examining "damaged areas over large regions, hence missing the details necessary to capture the impact of disasters on single unit houses or small areas". Or are the authors suggesting that their remote-sensing work, by contrast, is able to isolate individual buildings? If so, they are missing a significant body of academic literature that examines parcel-by-parcel damages from hurricanes. For example (and works related):

    - Deyle, R. E., Chapin, T. S., & Baker, E. J. (2008). The proof of the planning is in the platting: An evaluation of Florida's hurricane exposure mitigation planning mandate. *Journal of the American Planning Association*, *74*(3), 349-370.

    - Highfield, W. E., Peacock, W. G., & Van Zandt, S. (2014). Mitigation planning: Why hazard exposure, structural vulnerability, and social vulnerability matter. *Journal of Planning Education and Research*, *34*(3), 287-300.

    - Hamideh, S., Peacock, W. G., & Van Zandt, S. (2018). Housing Recovery after Disasters: Primary versus Seasonal/Vacation Housing Markets in Coastal Communities. *Natural Hazards Review*.

- L65 – Revisit this sentence? The deaths of 51 people did not leave homes without power – just needs a grammatical fix.

- L70 – "how the locals are dealing with these trends" – rework this sentence for better precision.

- L74 – Statement needs references.

- L94 – Saying that "permanent water bodies are excluded" here confuses a later calculation of distance to permanent water bodies. Suggest rewriting this to clarify that permanent water bodies are effectively set aside as their own category. They need to be differentiated from the flood extent, but they're still saved for another stage of the analysis.

- Section 2, more generally – The methods section sits as a big technical block, relative to the rest of the manuscript, as though a tool-based remote-sensing exercise has been jammed into an analysis that is otherwise conceptually straightforward. (That is, what was the total exposure of properties that ended up inside the flood footprint of Hurricane Florence near its landfall?) The authors might want to relegate much of the technical detail to a supplement, for readers who want to follow along, and only retain a stripped-back version in the main text.

  Furthermore – and more importantly – it's not clear from the text WHY the authors undertake the merger of satellite-based and FEMA data for flood extent. The merger forces the authors to spend significant page space trying to explain inconsistencies and uncertainties. That's fine, and important to do, but only if it's clear to the reader why this data synthesis (and remote-sensing exercise) is necessary in the first place. There is a sense that the authors are working with two papers here: one on the data-synthesis approach they describe, and another on the property stock under the flood footprint they derive. At the moment, these two elements are more competitive than mutually supportive.

- P11, LX00 [the line numbers get cut off once there are more than two digits, so I've switched convention] – Could the authors clarify here whether "total area" of the properties affected is the total taxed area of the building? Or the simple, plan-view physical footprint?

- P12, LX07 – first sentence is ungrammatical (and unclear as a result). Given those numbers (and the authors' explanation in that paragraph) FEMA's extent is by far more complete – which further begs the question, described above, about the utility of the satellite method here? Given the limitations to the image analysis the authors lay out, how confident are the they in the $3.3B that the satellite identified but FEMA did not?

  Does that render the satellite aspect of this work unnecessary (and a confusing addition), leaving the authors able to focus on FEMA's flood footprint and the property data?

- P13, LX35 – Why use Zillow for a median house price and not the ATTOM property dataset (and thus apples to apples)?

- P13, L55 – Suggest that this explanation of the expanding bull's-eye effect needs to go up to the beginning of the document, at first mention (around P5, LX30).

- **Section 3.3 – There is a significant issue with the analysis here that the authors will need to explain or address.** At P9, L38, the authors state, "Beside property values, the database also contains the year when each property was built, which we use for our expanding bull's-eye effect analysis."

  Information for year built is useful, but it is not a time-series. Unless I missed it (and apologies, if so), the authors do not say when their property data were compiled – but it looks like the dataset is housing stock as of 2018. In that case, they have a current snapshot of stock, not a continuous record of stock through time (i.e., annual records of all properties). That

means that the dataset will be inherently skewed toward newer properties, as old buildings get replaced. (See the same issue in Armstrong, S. B., Lazarus, E. D., Limber, P. W., Goldstein, E. B., Thorpe, C., & Ballinger, R. C. (2016). Indications of a positive feedback between coastal development and beach nourishment. Earth's Future, 4(12), 626-635.)

To demonstrate their bull's-eye effect, Ashley & Strader (2016) work with a semi-empirical spatio-temporal model of housing stock in tornado zones over time. Year-built records are not the same. Unless the authors can imagine a way to overcome this limitation in their analysis, they may not have the information they need to actually measure an expanding bull's-eye. The housing stock in their area of interest has certainly grown dramatically over time – the bull's eye is evident from space – but their properties dataset can only capture it indirectly.

- P15 – Conclusions section is overlong, I think, and can be distilled more succinctly into key points.

---

## Author Comment (AC1) · 23 Oct 2019

Dear Reviewer,

thanks for the suggestions for improving our manuscript. We hop you will find the revised version improved and suitable for publication.

Regards, M. Tedesco and co-authors
* * *

---

## Author Comment (AC2) · 26 Oct 2019

Dear Editor,

I am pleased to submit a revised version of the paper entitled "Exposure of properties to the 2018 Hurricane Florence flooding: an expanding bull's-eye perspective" for consideration on the Special Issue on Hydroclimatic extremes and impacts at catchment to regional scales, on the journal Natural Hazards and Earth System Sciences on behalf of co-authors and myself.

In our study, we aim at quantifying what was the total value and area of properties exposed to the flooding associated with Hurricane Florence. We first generate a map of the maximum flood extent from the combination of the extent produced by FEMA obtained from the interpolation of high water marks and flooded areas obtained by means of spaceborne radar remote sensing. Such map is, then, used for estimating the value and area of properties exposed to flooding and the distance of such properties from permanent water bodies. Lastly, we study and quantify how the urban development over the past years and decades over the regions flooded by Hurricane Florence might have impacted the exposure of properties and population to present-day storms and floods, to account what colleagues are starting to address as the "expanding bull's-eye effect" in which ''targets" of geophysical hazards, such
as people and their built environments, are enlarging as populations grow and spread.

We appreciate the input and suggestions provided by reviewers and we have done our best to include them. In this regard, we have moved a relatively large section of the manuscript dealing with radar data in the Supplementary Material, we have shortened the Conclusions and have (hopefully) improve the style, as requested. In this regard, we believe that the combination of radar data and FEMA maps is an important aspect of our paper as it shows the complementary nature of these datasets. In some cases, we were not able to satisfy the requests by reviewers, especially when it comes to expressing the exposed property value in terms of percentage of taxes or other metrics. We are currently working on this aspect and it takes some time before this can be done. We think this is outside of the scope of our manuscript and reporting absolute values will allow our results to be compared with others reported in the literature.

We hope you will find the revised version improved. We are attaching two versions: one showing the changes and another one without showing them. We hope this would help reviewers and the editor in reading the revised version.

Sincerely,
M. Tedesco and co-authors
10/25/2019

**Anonymous Referee #1**

1. General: The writing style should be more formal and should rely more on active verbs. In general, the manuscript should be edited down to simplify the sentence structure. In addition, there are many errors of punctuation and grammar that are distracting.

*R: We have revised the document and we hope the reviewer finds the new version improved in style.*

2. Findings: The inadequacy of FEMA flood mapping should be reinforced and contextualized. For instance, how can this method augment or support alternative determinations of spatial risk in the housing market? I would like to see some analysis of the method and findings within matters of local and federal policy. For instance, could this method compliment Hazus? Maybe or maybe not. At its core, this is about land use and zoning and you should be explicit about this.

*R. We agree that this is an important point and consideration. We have added a section to page 11 in which we briefly identify the potential complementary nature of the radar and FEMA extents as a way to more holistically capture the event impact areas. Additionally, we agree that the specific type of land that is captured as flooding by the radar might have to be considered in regards to any potential policy implications. It is beyond the scope of this paper to actually test this, but we do note on page 11 that more research would be necessary to fully understand the utility of the method moving forward.*

3. Line 45: There is also some counter- and supporting-evidence on the horizon that is worth reading:

Gibson, M., J. Mullins, and A. Hill 2017. "Climate Change, Flood Risk, and Property Values: Evidence from New York City". Working paper, University of Massachusetts-Amherst.

Murfin, J. and M. Spiegel 2018. "Is the Risk of Sea Level Capitalized in Residential Real Estate?" Review of Financial Studies Climate Finance Call, Conditionally Accepted.

R: We thank the reviewer for the suggestion. We have searched for the material on the web and we were not able to locate a published version of the first paper and the copy we found of the second one mentions not to cite it as it is an incomplete version. We will definitely read and include this paper in our future studies.

4. Present Value: When citing valuations, it is always good practice to cite the associated dollar year.

*R: We thank the reviewer for this suggestion and we have added a sentence in the introduction specifying that we use 2018 as a dollar year.*

5. Lines 58-70: This should be significantly edited-down. I think it is also useful to highlight the economic distinctions between stresses and shocks because they have different theoretical pricing elasticities.

*R.The distinction has been added to pages 4-5. We now distinguish between market stressors and shocks as they pertain to the specific types of flooding events, the potential impacts on elasticity, and the identification of Hurricane Florence as our focus within that context.*

6. Line 86: "Of particular importance to the recent market response is the fact that increased probability seems to be an important driving force." A "fact" and "seems" are

contradictory language here.

*R : we replaced "sems to be"with "is"*

7. Lines 103-108: It is critical that you use the precise language of exposure and sensi-tivity within the conceptual framing of vulnerability. Much of your pivot argument around urban development is fundamentally about exposure that has merit independent of climate attributed phenomena.

*R. Thank you for this comment and we completely agree. We have attempted to make that clear in the most recent version of the paper with statements like "It is important to note that much of the vulnerability associated with building development in these areas should be considered independent of climate change to this point. However, moving forward, these types of storms are expected to increase in intensity and the link between climate change and potential exposure is likely to be tied more closely together. In fact, we are already seeing these trends as they relate to tidal flooding events and one might expect that the low probability-larger storms are likely to become more empirically linked to our changing climate as well".*

8. Dates: We only need years–not days and months (unless you are referencing SAR).

*R: We apologize with the reviewer but it is not clear what he/she is asking. We use day, month and year for describing the series of events that characterized the Florence hurricane. Having days of the month helps to put things in the context of the SAR data and to explain why the SAR data has missed the maximum flood extent. Again, we are not sure what the suggestion is.*

9. Metric System: I would recommend converting to the metric system.

*R: We are now using the metric system throughout the entire document.*

10. Line 130: I would go ahead and define the "bulls-eye."

*R: We thank the reviewer and we have defined the "bull's-eye" as the eye of the storm.*

11. Line 138, et seq.: Please define acronyms in their first usage.

*R: We have done this, thanks for the suggestion*

12. Section 2.3: Are there other data sources from which you can triangulate and vali-date assessed value with market value to normalize across the distribution to account for varying appraisal methods and time limitations? I'm not sure I understand how this relates to the Zillow data that you later cite.

R. We make use of both ATTOM and Zillow data because they are measuring 2 qualitatively different values. The ATTOM data comes directly from the county Assessor's Office and includes the most accurate version of the Assessment Value, per the county's specific approach to quantifying the local tax base. In addition to that, we were interested in the potential Market Value impact. For that data, we turned to Zillow's aggregated Zestimate. The Zillow data is an automated market value product which should better represent the current market value of any home, or group of homes, relative to the current state of the respective location. As an example of the quality of the metric, see the image below which documents the longitudinal tracking of the Zillow market value estimate against the Case-Schiller Actual Home Value Index.

https://wp.zillowstatic.com/blogs-legacy/1/files/2010/04/image002.png

On the other hand, the Assessment data simply provides a tool from which the local government can understand its tax-base… even if that number is not as accurate as the actual market value.

13. Page 14, Line 369-70: "An explosion in new properties occurred between 1950 and 2000 (Figure 370 9c), likely as a consequence of the economic stimulus following World War II." This is not a sound association over the cited length of time. Economic research would suggest that "half-back" population shifts associated with these areas being low-cost retirement areas for high-cost Northeaster residents is the primary driver of coastal development patterns here.

*R: We thank the reviewer for this suggestion. We admit that our phrase, the way it is written, excludes this possibility. In view of this, we have decided to remove the sentence, also to improve the readability of the manuscript.*

14. Page 5, Line 125: " no study, to our knowledge, has focused on the impact of urban development on the property exposed to Hurricane Florence." I suppose this is true, but the real contribution is the time-space distribution effects on absolute exposure– independent of climate change attributed events. This exposure has happened and exists whether or not climate change impacts are measured or not.

*R: We agree with the reviewer. We are not sure what the reviewer is asking and we did not change the sentence as we specifically focus on Hurricane Florence in this article through the use of the flooded areas for estimating the financial exposure.*

15. Page 14: If you could find a weighted average assessment tax, it would be helpful to contextual how important these properties are to local tax bases (county-level). The raw numbers are hard to contextualize.

R. We report in the current version of the paper that the total losses reported amount to about 10% of the total property value across the entire study region. Additionally, due to the inconsistency associated with the way tax rates are created, the market value was seen as a more reliable indicator of proportional impact. We do have students working on the compilation of county level tax rates so that we can look into this in the future.

16. Page 16: When you cite the increase in valuation, my first question is: so what? You really need to another metric for contextualizing exposure. Citing 1940s valuation is methodologically not particularly sound anyway. Again, I recommend looking at local tax base and operating in percentage share.

R. Our data are cross-sectional and represent the 2018 housing stock, but when looking at the impact on total value over time we find that there was little deviation from the 10% affected rate.We thank the reviewer for this suggestion and, as mentioned above, we have started to look into this. However, this might take time and we prefer to keep the absolute values in the current version of the manuscript. Moreover, many of the estimates in the literature and media are reported in absolute $ , making our estimates comparable to those.

17. Overall: This paper needs to be significantly edited and simplified. The work is largely solid, but it meanders too much. The connections to climate change are somewhat disconnected as a matter of attribution. The value is the observation method and the context of expanding exposure. The science appears to be well-grounded (beyond my competency to full evaluate), but the economics needs some work.

*R: We have revised the manuscript according to this and other suggestions by reviewers. We hope the reviewer will find the new version improved.*

**REVIEW – nhess-2019-209 (Tedesco *et al.*)**
This manuscript aims to quantify an "expanding bull's-eye effect" in the densely developed region of the US Atlantic Coast hit hardest by Hurricane Florence in 2018. Overall, the concept for the article is engaging, and represents a useful analysis of how exposure of physical property to coastal flood risk has changed over time.
I have some broad suggestions (organised by line number) for the authors that I hope might improve the manuscript. I do think there is one major issue (see remarks on Section 3.3, below) that will need to be resolved before this work is ready for publication.

- L45 – the "now problem" phrase is cumbersome and seems unnecessary. Consequences of flooding from sea-level rise has been a fixture of IPCC reports, etc – I think the authors should simplify/clarify this rhetorical line or cut it for a straighter delivery.
- *R: We removed that sentence*
- L50 – I don't understand the counterpoint in this sentence/paragraph. This work is also examining "damaged areas over large regions, hence missing the details necessary to capture the impact of disasters on single unit houses or small areas". Or are the authors suggesting that their remote- sensing work, by contrast, is able to isolate individual buildings? If so, they are missing a significant body of academic literature that examines parcel- by-parcel damages from hurricanes. For example (and works related):

o Deyle, R. E., Chapin, T. S., & Baker, E. J. (2008). The proof of the planning is in the platting: An evaluation of Florida's hurricane exposure mitigation planning mandate. *Journal of the American Planning Association*, *74*(3), 349-370.
o Highfield, W. E., Peacock, W. G., & Van Zandt, S. (2014). Mitigation planning: Why hazard exposure, structural vulnerability, and social vulnerability matter. *Journal of Planning Education and Research*, *34*(3), 287-300.
o Hamideh, S., Peacock, W. G., & Van Zandt, S. (2018). Housing Recovery after Disasters: Primary versus Seasonal/Vacation Housing Markets in Coastal Communities. *Natural Hazards Review*.
*R: We thank the reviewer for the suggestion on the papers and we do realize that our sentence might suggest to exclude some existing work. We have, therefore, removed this sentence.*

- L65 – Revisit this sentence? The deaths of 51 people did not leave homes without power – just needs a grammatical fix.
- *R: Done, thanks !*
- L70 – "how the locals are dealing with these trends" – rework this sentence for better precision.
- *R: This sentence has been removed to make the paper more readable, as requested by both reviewers.*
- L74 – Statement needs references.
- *R: We have added a reference as requested.*
- L94 – Saying that "permanent water bodies are excluded" here confuses a later calculation of distance to permanent water bodies. Suggest rewriting this to clarify that permanent water bodies are effectively set aside as their own category. They need to be differentiated from the flood extent, but they're still saved for another stage of the analysis.

- *R: Thanks ! We have revised the sentence to clarify this point.*

Section 2, more generally – The methods section sits as a big technical block, relative to the rest of the manuscript, as though a tool-based remote-sensing exercise has been jammed into an analysis that is otherwise conceptually straightforward. (That is, what was the total exposure of properties that ended up inside the flood footprint of Hurricane Florence near its landfall?) The authors might want to relegate much of the technical detail to a supplement, for readers who want to follow along, and only retain a stripped-back version in the main text.

*R: We moved a large portion of the technical description concerning SAR in a Supplementary section. We thank the reviewer for this suggestion as the manuscript readability has been considerably improved.*

Furthermore – and more importantly – it's not clear from the text WHY the authors undertake the merger of satellite-based and FEMA data for flood extent. The merger forces the authors to spend significant page space trying to explain inconsistencies and uncertainties. That's fine, and important to do, but only if it's clear to the reader why this data synthesis (and remote-sensing exercise) is necessary in the first place. There is a sense that the authors are working with two papers here: one on the data- synthesis approach they describe, and another on the property stock under the flood footprint they derive. At the moment, these two elements are more competitive than mutually supportive.

*R: We understand that, in its original form, the role of SAR and FEMA data might have been perceived as "competing". As pointed out by reviewers, this might have also due to the larger role played by the description of the SAR methods in the original manuscript. We have, now, moved a large portion of the SAR technical section into a supplementary material and the manuscript is, hopefully, more balanced. In our manuscript we show that the SAR approach might miss the maximum flood extent because of the acquisition time and that the FEMA approach might miss some of the flooded areas captured by SAR, because of potential intrinsic limitations of the FEMA tool. In view of this, we think that the two approaches are actually complimentary (rather than competitive). We also think that adding SAR information in our manuscript might promote knowledge of alternative techniques to the ones developed by FEMA for those readers who do not have a remote sensing background.*

- P11, LX00 [the line numbers get cut off once there are more than two digits, so I've switched convention] – Could the authors clarify here whether "total area" of the properties affected is the total taxed area of the building? Or the simple, plan-view physical footprint?
- *R: We added it is the physical footprint*
- P12, LX07 – first sentence is ungrammatical (and unclear as a result). Given those numbers (and the authors' explanation in that paragraph) FEMA's extent is by far more complete – which further begs the question, described above, about the utility of the satellite method here? Given the limitations to the image analysis the authors lay out, how confident are the they in the $3.3B that the satellite identified but FEMA did not?

Does that render the satellite aspect of this work unnecessary (and a confusing addition), leaving the authors able to focus on FEMA's flood footprint and the property data?

*R: We thank the reviewer for this suggestion. We have re-written the sentence. We think that showing the values missed by the FEMA approach, despite being relatively small, shows that the two methods are complementary. As we point out in our manuscript, the FEMA method might miss some of the flooded areas because of issues related to interpolation and/or lack of spatial coverage (either related to missing data collected on the ground or to the model's domain). These points are not missed by the SAR. The combination of the two methods, therefore, increases our confidence in the fact that more flooded areas will be captured. Given the issues at stake, we would like to keep the estimates from both methods. The confidence of the SAR method comes from the comparison of the two water masks obtained from the two methods, giving us high confidence on the estimated values.*

- P13, LX35 – Why use Zillow for a median house price and not the ATTOM property dataset (and thus apples to apples)?
- *R: We have replaced the Zillow esrimates with those from our database. We originally used Zillow as that it is easily accessible to readers but we agree with the reviewer that this is a better option.*
- P13, L55 – Suggest that this explanation of the expanding bull's-eye effect needs to go up to the beginning of the document, at first mention (around P5, LX30).
- *R: Following also another reviewer's suggestion, we have added that in the Introduction section.*
- **Section 3.3 – There is a significant issue with the analysis here that the authors will need to explain or address.** At P9, L38, the authors state, "Beside property values, the database also contains the year when each property was built, which we use for our expanding bull's-eye effect analysis."

Information for year built is useful, but it is not a time-series. Unless I missed it (and apologies, if so), the authors do not say when their property data were compiled – but it looks like the dataset is housing stock as of 2018. In that case, they have a current snapshot of stock, not a continuous record of stock through time (i.e., annual records of all properties). That means that the dataset will be inherently skewed toward newer properties, as old buildings get replaced. (See the same issue in Armstrong, S. B., Lazarus, E. D., Limber, P. W., Goldstein, E. B., Thorpe, C., & Ballinger, R. C. (2016). Indications of a positive feedback between coastal development and beach nourishment. Earth's Future, 4(12), 626-635.)

To demonstrate their bull's-eye effect, Ashley & Strader (2016) work with a semi-empirical spatio-temporal model of housing stock in tornado zones over time. Year-built records are not the same. Unless the authors can imagine a way to overcome this limitation in their analysis, they may not have the information they need to actually measure an expanding bull's- eye. The housing stock in their area of interest has certainly grown dramatically over time – the bull's eye is evident from space – but their properties dataset can only capture it indirectly.

*R: To clarify: we use the year when the property was built with the property value computed for the year 2018 to calculate the potential exposure of properties to the*

*flood generated by hurricane Florence. Said this, we acknowledge that the reviewer point is important. The bulls-eye effect is the product or our indirect visualization of the change over time based solely on the data from 2018. As such, we are unable to employ the same spatio-temporal due to the cross-sectional nature of the data. Moving forward, it would be very interesting to incorporate such a technique with temporally varying data, especially given the fact that storms and flooding could dramatically change the landscape by completely removing previous development from the 2018 housing stock.*

• P15 – Conclusions section is overlong, I think, and can be distilled more succinctly into key points.

*R: WE have revised and reduced the length of the conclusions.*

[revised manuscript text omitted]

It is, therefore, crucial to study how the urban development over the past years or decades might have impacted the exposure of properties and population to present-day storms and floods. For example, one of the most devastating hurricanes over the Carolinas before Florence was Hurricane Hugo, reaching the Carolinas on 10 September 1989, with winds up to 260 Km/hour and a total estimated damage of $9.45 billion (in 1989 USD, equivalent to ~ $19B of 2018 USD) and 60 fatalities. Unlike 1989, we have today improved observational and modeling tools that allow us to better estimate the maximum flood extent, a key parameter needed to estimate the potential exposure to damage of properties and other infrastructures. From a modeling point of view, hydrological and hydrodynamic models, in conjunction with improved digital elevation models and the ingestion of gage observation or observation of high water marks, offer the opportunity to generate estimates of maximum flood extent (FEMA, 2019).

**1.2 Purpose of this study**

Despite recent studies have started to focus on the spatio-temporal variability of property values and human settlements in hurricane-prone areas (e.g., Huang et al., 2019) and on the market responses to increases in observed flooding events (e.g., McAlpine and Porter, 2019; Keenan et. al, 2018), no study, to our knowledge, has focused on the impact of urban development on the property exposed to Hurricane Florence. Addressing this point is crucial to account for those impacts related to the choices that our society makes to continue the expansion of urban areas and that have been addressed by experts as the "bull's-eye expanding effect" (Ashley and Strader, 2018), in which "targets" of geophysical hazards, such as people and their built environments, are enlarging as populations grow and spread. We

Marco T 10/21/19 17:25

Marco T 10/21/19 17:25

Marco T 10/21/19 17:25

Marco T 10/21/19 17:26

Marco T 10/21/19 17:26

Marco T 10/21/19 17:26

Marco T 10/21/19 17:30

Marco T 10/21/19 17:30

Marco T 10/21/19 17:30

[revised manuscript text omitted]

The complementary nature of the FEMA and SAR-based approaches is helpful to provide more robust maximum flood extent maps than the ones that can be obtained from the two approaches separately. If this is proven to be a reliable method for extent identification, one could easily see it being incorporated in to standard techniques used to identify damage estimates (for instance, into Hazus models). That being said, it is also clear that the radar tends to pick up flooding in agricultural/less populated spaces and that distinction is important and worth spending more time confirming. Given these considerations, for this study we merge the FEMA and Sentinel-1 flood extent maps to generate a maximum composite flood extent map that will be used to assess the property exposure to Hurricane Florence flooding. We will refer to this dataset simply as the "maximum flood extent" in the remaining sections of the manuscript.

**3.2 Exposure of property to Hurricane Florence flooding**

Figure 5 shows the spatial distribution of the properties within our database overlaid with an image of the eye of Hurricane Florence when it made landfall. Our results indicate that the total area of properties affected by the maximum flood extent water was 70,964,700 m$^2$ (e.g., physical footprint) being 17.55 % of the total area within our database. When considering only the flood extent estimated by Sentinel-1, the total area of properties affected by the estimated flood extent reduces to 3.2 %, corresponding to 12,939,432 m$^2$. In order, to quantify potential biases associated with co-registration issues or resampling procedures, we computed the number of properties exposed to the extent of our

Marco T 10/22/19 05:38

Marco T 10/21/19 19:14

Marco T 10/21/19 19:14

Marco T 10/21/19 19:17

Marco T 10/21/19 19:17

Marco T 10/22/19 05:38

Marco Tedesco 9/26/19 01:05

Marco T 10/21/19 19:24

permanent body water dataset. Our analysis shows that less than 0.2 % of properties was overlapping with the permanent body waters. Consequently, we removed these properties from our analysis.

[revised manuscript text omitted]

Marco Tedesco 10/9/19 15:25

---

## Referee Report (RR1)

**REVIEW – NHESS-2019-209 (v2)**

I was glad to have a second look at this manuscript following a revision.

A further, minor remark:

P5,6 – *"Addressing this point is crucial to account for those impacts related to the choices that our society makes to continue the expansion of urban areas and that have been addressed by experts as the "bull's-eye expanding effect" (Ashley and Strader, 2018), in which ''targets'' of geophysical hazards, such as people and their built environments, are enlarging as populations grow and spread. We use the term "bull's-eye" to define the eye or center of a storm."* – Given the Introduction, seems a strange reorientation of the bull's eye terminology. Instead of the target, makes it sound like the bull's-eye refers to the dart – as though the argument here will be that storm size is increasing (and not an elaboration of the premise that hazard exposure is increasing). Not sure that this is the case the authors intend to frame?

--

---

## Editor Decision (ED1)

Dear authors

The new version of the manuscript has improved. Nevertheless, there are some points that requires more effort or a better explanation

1) The presentation: the presentation of the paper has improved from the first version but in my opinion need more effort. In the new version remain some sentences difficult to understand:  e.g. page 12 line 21: "As mentioned in the Introduction, the exposure to floods and other extreme events depends not only on the geophysical hazard but also on how urban growth and infrastructures have been, are and will be evolving in the areas at risk". Another important error of presentation is that figure 12 is not cited in the text. Please review all the manuscript taking care of this questions.

2) The critical point of reviewer 2 have not been addressed properly. "*Section 3.3 – There is a significant issue with the analysis here that the authors will need to explain or address. At P9, L38, the authors state, "Beside property values, the database also contains the year when each property was built, which we use for our expanding bull's-eye effect analysis." Information for year built is useful, but it is not a time-series. Unless I missed it (and apologies, if so), the authors do not say when their property data were compiled – but it looks like the dataset is housing stock as of 2018. In that case, they have a current snapshot of stock, not a continuous record of stock through time (i.e., annual records of all properties). That means that the dataset will be inherently skewed toward newer properties, as old buildings get replaced. (See the same issue in Armstrong, S. B., Lazarus, E. D., Limber, P. W., Goldstein, E. B., Thorpe, C., & Ballinger, R. C. (2016). Indications of a positive feedback between coastal development and beach nourishment. Earth's Future, 4(12), 626-635.) To demonstrate their bull's-eye effect, Ashley & Strader (2016) work with a semiempirical spatio-temporal model of housing stock in tornado zones over time. Year-built records are not the same. Unless the authors can imagine a way to overcome this limitation in their analysis, they may not have the information they need to actually measure an expanding bull's- eye. The housing stock in their area of interest has certainly grown dramatically over time – the bull's eye is evident from space – but their properties dataset can only capture it indirectly*".
In your reply you accept the limitations of the data but you must to include some comment on this limitation of your data/analysis in the text.

3) Please address the minor remark of the referee:  P5,6 – "Addressing this point is crucial to account for those impacts related to the choices that our society makes to continue the expansion of urban areas and that have been addressed by experts as the "bull's-eye expanding effect" (Ashley and Strader, 2018), in which ''targets'' of geophysical hazards, such as people and their built environments, are enlarging as populations grow and spread. We use the term "bull's-eye" to define the eye or center of a storm." – Given the Introduction, seems a strange reorientation of the bull's eye terminology. Instead of the target, makes it sound like the bull's-eye refers to the dart – as though the argument here will be that storm size is increasing (and not an elaboration of the premise that hazard exposure is increasing). Not sure that this is the case the authors intend to frame?

---

## Author Response (AR2)

**Reply to reviewers**

1) The presentation: the presentation of the paper has improved from the first version but in my opinion need more effort. In the new version remain some sentences difficult to understand: e.g. page 12 line 21: "As mentioned in the Introduction, the exposure to floods and other extreme events depends not only on the geophysical hazard but also on how urban growth and infrastructures have been, are and will be evolving in the areas at risk". Another important error of presentation is that figure 12 is not cited in the text. Please review all the manuscript taking care of this questions.

*R: We thanks the Editor for this comment. We have revised the entire manuscript for clarity and have added the reference to Figure 12 (it was erroneously cited as Figure 13).*

2) The critical point of reviewer 2 have not been addressed properly. "Section 3.3 – _There is a significant issue with the analysis here that the authors will need to explain or address. At P9, L38, the authors state, "Beside property values, the database also c_o_n_t_a_i_n_s_ _t_h_e_ _y_e_a_r_ _w_h_e_n_ _e_a_c_h_ _p_r_o_p_e_r_t_y_ _w_a_s_ _b_u_i_l_t_,_ _w_h_i_c_h_ _w_e_ _u_s_e_ _f_o_r_ _o_u_r_ _e_x_p_a_n_d_i_n_g_ _b_u_l_l_'s_-eye effect analysis." Information for year built is useful, but it is not a time-series. Unless I missed it (and apologies, if so), the authors do not say when their property data were compiled – _but it looks like the dataset is housing stock as of 2018. In that case, they have a current snapshot of stock, not a continuous record of stock through time (i.e., annual records of all properties). That means that the dataset will be inherently skewed toward newer properties, as old buildings get replaced. (See the same issue in Armstrong, S. B., Lazarus, E. D., Limber, P. W., Goldstein, E. B., Thorpe, C., & Ballinger, R. C. (2016). Indications of a positive feedback between coastal development and beach nourishment. Earth's Future, 4(12), 626-635.) To demonstrate their bull's-eye effect, Ashley & Strader (2016) work with a semiempirical spatio-temporal model of housing stock in tornado zones over time. Year-built records are not the same. Unless the authors can imagine a way to overcome this limitation in their analysis, they may not have the information they need to actually measure an expanding bull's- eye. The housing stock in their area of interest has certainly grown dramatically over time – _the bull's eye is evident from space – _but their properties dataset can only capture it indirectly".

In your reply you accept the limitations of the data but you must to include some comment on this limitation of your data/analysis in the text.

*R: We have clarified how our results might differ from the original work and addressed the limitations of our approach.*

3) Please address the minor remark of the referee: P5,6 – _"Addressing this point is crucial to account for those impacts related to the choices that our society makes to continue the expansion of urban areas and that have been addressed by experts as the "b_u_l_l_'s_-e_y_e_ _e_x_p_a_n_d_i_n_g_ _e_f_f_e_c_t_" _(A_s_h_l_e_y_ _a_n_d_ _S_t_r_a_d_e_r_,_ _2_0_1_8_)_,_ _i_n_ _w_h_i_c_h_ _''t_a_r_g_e_t_s_" _o_f_ _geophysical hazards, such as people and their built environments, are enlarging as populations_ _g_r_o_w_ _a_n_d_ _s_p_r_e_a_d_._ _W_e_ _u_s_e_ _t_h_e_ _t_e_r_m_ _"b_u_l_l_'s_-e_y_e_" _t_o_ _d_e_f_i_n_e_ _t_h_e_ _e_y_e_ _o_r_ _c_e_n_t_e_r_ _of a storm." – _Given the Introduction, seems a strange reorientation of the bull's eye terminology. Instead of the target, makes it sound like the bull's-eye refers to the dart – _as though the argument here will be that storm size is increasing (and not an elaboration of the premise that hazard exposure is increasing). Not sure that this is the case the authors intend to frame?

P5,6 – "Addressing this point is crucial to account for those impacts related to the choices that our society makes to continue the expansion of urban areas and that have been addressed by experts as the "bull's-eye expanding effect" (Ashley and

Strader, 2018), in which ''targets" of geophysical hazards, such as people and their built environments, are enlarging as populations grow and spread. We use the term "bull's-eye" to define the eye or center of

a storm." – Given the Introduction, seems a strange reorientation of the bull's eye terminology. Instead of the target, makes it sound like the bull's-eye refers to the dart – as though the argument here will be that storm size is increasing (and not an elaboration of the premise that hazard exposure is increasing). Not sure that this is the case the authors intend to frame?

**R: We thank the editor for the suggestion. We have re-worded that part, paying attention to specify the fact that the hazard exposure is increasing as follows:**

Original text:
Addressing this point is crucial to account for those impacts related to the choices that our society makes to continue the expansion of urban areas and that have been addressed by experts as the "bull's-eye expanding effect" (Ashley and
Strader, 2018), in which ''targets" of geophysical hazards, such as people and their built environments, are enlarging as populations grow and spread. We use the term "bull's-eye" to define the eye or center of a storm.

*Revised text:*
*Addressing this point is crucial to account for those impacts related to urban growth and the expansion of urban areas as addressed by experts when considering the so-called the "bull's-eye expanding effect" (Ashley and Strader, 2018), in which ''targets" of geophysical hazards, such as people and their built environments, are enlarging as populations grow and spread. In this case, the bull's eye expansion does not refer to the increased storm size but rather to the area where the impact of the geophysical hazard is occurring, expanding because of the urbanization process over the past decades.*

**Exposure of real estate properties to the 2018 Hurricane Florence flooding**

Marco Tedesco[1], Steven McAlpine[2] and Jeremy R. Porter[3, 4]

1) Lamont-Doherty Earth Observatory of the Columbia University
2) FirstStreet Foundation
3) City University of New York - Quantitative Methods in the Social Sciences
4) Columbia University Medical Center - Environmental Health Sciences

*Correspondence to*: M.Tedesco (mtedesco@ldeo.columbia.edu)

**Abstract.** Quantifying the potential exposure of property to damages associated with storm surges, extreme weather, and hurricanes is fundamental to developing frameworks that can be used to conceive and implement mitigation plans as well as support urban development that accounts for such events. In this study, we aim at quantifying the total value and area of properties exposed to the flooding associated with Hurricane Florence that occurred in September 2018. To this aim, we implement an approach for the identification of affected areas by generating a map of the maximum flood extent obtained from a combination of the flood extent produced by the Federal Emergency Management Agency's (FEMA) water marks with those obtained from spaceborne radar remote sensing data. The use of radar in the creation of the flood extent allows for those properties commonly missed by FEMA's interpolation methods, especially from pluvial/non-fluvial sources, and can be used in more accurately estimating the exposure and market-value of properties to event-specific flooding. Lastly, we study and quantify how the urban development over the past decades in the regions flooded by Hurricane Florence might have impacted the exposure of properties to present-day storms and floods. This approach is conceptually similar to what experts are addressing as the "expanding bull's-eye effect" in which ''targets'' of geophysical hazards, such as people and their built environments, are enlarging as populations grow and spread. Our results indicate that the total value of property exposed to flood during Hurricane Florence was $52B (in 2018 USD), with this value increasing from ~ $10B at the beginning of the past century to the final amount based on the expansion of number of properties
* * *
Marco T 2/3/20 11:29

Marco T 2/3/20 11:40

Marco T 2/3/20 11:40

Marco T 2/3/20 11:40

Marco T 2/3/20 11:40

Marco T 2/3/20 11:41

Marco T 2/3/20 11:41

Marco T 2/3/20 11:41

Marco T 2/3/20 11:41

Marco T 2/3/20 11:42

Marco T 2/3/20 11:42

Marco T 2/3/20 11:43

[revised manuscript text omitted]

The complementary nature of the FEMA and SAR-based approaches is helpful to provide more robust maximum flood extent maps than the ones that can be obtained from the two approaches separately. If this is proven to be a reliable method for extent identification, one could easily see it being incorporated in to standard techniques used to identify

damage estimates (for instance, into Hazus models).  That being said, it is also clear that the radar tends to pick up flooding in agricultural/less populated spaces and that distinction is important and worth spending more time confirming.

| Page 12: [5] Deleted | Marco T | 2/10/20 06:41 |

In order, to investigate the impact of the expanding bull's-eye effect on the property exposed to the flooding of Hurricane Florence,

indeed, that climate change is and will be influencing the frequency and strength of storms and floods, it is also true that the exposure of anthropogenic structures and lives is a function of urbanization factors such as, for example, the building of new properties in proximity of the coast and of body waters. In this context, it becomes crucial to understand and quantify how urban development has impacted the

5   exposure of properties and population to present-day storms and floods. For example, one of the most devastating hurricanes over the same region before Florence was Hurricane Hugo, reaching the Carolinas on 10 September 1989, with winds up to 260 Km/hour and a total estimated damage of $9.45 billion (in 1989 USD, equivalent to ~ $19B of 2018 USD) and 60 fatalities. Unlike 1989, we have today improved observational and modeling tools that allow us to better estimate the maximum flood extent, a

10   key parameter needed to estimate the potential exposure to damage of properties and other infrastructures. From a modeling point of view, hydrological and hydrodynamic models, in conjunction with improved digital elevation models and the ingestion of gage observation or observation of high water marks, offer the opportunity to generate estimates of maximum flood extent (FEMA, 2019).

We aim at understanding the usefulness of remotely sensed satellite data as a method for the

15   identification of impacted areas and for delineating the maximum flood extent. Specifically, we report results concerning the mapping of the flood extent associated with Hurricane Florence estimated from SAR data and compare such extent with the maximum flood extent provided by FEMA. From that exposure, we are able to quantify the property value and total area exposed to Hurricane Florence by combining the flood extent coverage with a database containing publicly available property value

20   attributes. Despite recent studies have started to focus on the spatio-temporal variability of property values and human settlements in hurricane-prone areas (e.g., Huang et al., 2019) and on the market responses to increases in observed flooding events (e.g., McAlpine and Porter, 2019; Keenan et. al, 2018), no study, to our knowledge, has focused on the impact of urban growth on the property exposed to Hurricane Florence. Addressing this point is crucial to account for those impacts related to the

25   choices that our society makes to continue the expansion of urban areas and that have been addressed by experts as the "bull's-eye expanding effect" (Ashley and Strader, 2018), in which "targets" of geophysical hazards, such as people and their built environments, are enlarging as populations grow and spread. The term "bull's-eye" is here used to define the eye or center of a storm. Our approach is

Marco T 2/3/20 16:08

Marco T 2/3/20 16:08

Marco T 2/3/20 16:07

Marco T 2/3/20 16:09

Marco T 2/3/20 16:11

Marco T 2/3/20 16:11

Marco T 2/3/20 16:11

Marco T 2/3/20 16:11

Marco T 2/3/20 16:11

Marco T 2/3/20 16:11

Marco T 2/3/20 16:11

Marco T 2/3/20 16:12

Marco T 2/3/20 16:13

Marco T 2/3/20 12:01
**1.2 Purpose of this study** .

Marco T 2/3/20 12:01

Marco T 2/3/20 16:26

Marco T 2/3/20 16:26

[revised manuscript text omitted]

Marco T 2/3/20 16:37

Marco T 2/3/20 18:26

Marco T 2/3/20 18:29

Marco T 2/3/20 18:29

Marco T 2/3/20 18:29

Marco T 2/3/20 18:30

Marco T 2/3/20 18:31

Marco T 2/3/20 18:31

Marco T 2/3/20 18:30

Marco T 2/3/20 18:31

Marco T 2/3/20 18:32

Marco T 2/3/20 18:37

Marco T 2/3/20 18:37

backscattering coefficients (not shown here) indicates that the backscattering values recorded for those regions where flood was identified by the radar were relatively low (e.g., well below the threshold value and on the order of ~ -20 dB or below), indicating that those were, indeed, inundated areas.

Another factor complicating the comparison between Sentinel-1 and FEMA inundated regions regards the acquisition time of the radar images, which are collected before or after the time of the maximum water extent. Figure 3a shows the time series of the water height (mean sea level in meters) for the ocean tide gauge located in Wrightsville Beach, NC (id #8658163), where Hurricane Florence made landfall. Maximum water height was reached on the same day around 15:00 UTC. The image also shows the acquisition time of the Sentinel-1B (14 September, 2018, 11:15:05, UTC) and Sentinel-1A (14 September, 2018, 23:05:48, UTC) as vertical, dashed lines, indicating that such images were, indeed, acquired before and after the time when the water reached the maximum extent. River gages data also show that, because of the heavy precipitation, the maximum water discharge and gage heights inland occurred a few days after hurricane Florence made landfall. In this regard, Figures 3b and 3c show, respectively, the daily discharge (in cubic meters per hour) and daily gage height (in meters) recorded at the river gauge station of Lumberton, NC (34.6182° N, 79.0086° W), located about 150 km inland. The data shows the peak discharge and water heights late in the evening of 17 September, 2018. For this same area the radar data were collected when the tide gage recorded peak values, confirming the usefulness of this tool to capture flooding that might not have been captured by FEMA. As a further example, we show in Figure 4 the flooded areas detected by Sentinel-1 (blue filled regions) on 19 September, 2018 nearby Pasley, Duplin County, NC (34.7854° N, 77.9005° W) and a photograph of the same area collected on 18 September, 2018 by the NOAA Remote Sensing Division to support emergency response requirements (https://storms.ngs.noaa.gov/storms/florence/index.html#7/35.360/-77.820). The figure shows that most of the flooded areas identified within the NOAA photograph are properly captured by Sentinel-1, with differences between the two also due to the different acquisition times. For this area, the FEMA map does not indicate any flooding, confirming the complementary nature of the radar dataset.

Given these considerations, for this study we merge the FEMA and Sentinel-1 flood extent maps to generate a maximum composite flood extent map that will be used to assess the property exposure to

Marco T 2/3/20 18:42

Marco T 2/3/20 18:42

Marco T 2/3/20 18:42

Marco T 2/3/20 18:43

Marco T 2/3/20 18:44

Marco T 2/3/20 18:45

Marco T 2/3/20 18:45

Marco T 2/3/20 18:45

Marco T 2/3/20 18:46

Marco T 2/3/20 18:50

Marco T 2/3/20 18:50

Marco T 2/3/20 18:51

Marco T 2/3/20 18:52

Marco T 2/3/20 18:52

Marco T 2/3/20 18:52

Marco T 2/3/20 18:52

Marco T 2/10/20 05:57
Deleted: The complementary nature of the FEMA and SAR-based approaches is helpful to provide more robust maximum flood extent maps than the ones that can be obtained from the two approaches separately. If this is proven to be a reliable method for extent identification, one could easily see it being incorporated in to standard techniques used to identify damage estimates (for instance, into Hazus models).  That being said, it is also clear that the radar tends to pick up flooding in agricultural/less populated spaces and th ... [4]

Hurricane Florence flooding. We will refer to this dataset simply as the "maximum flood extent" in the remaining sections of the manuscript.

**3.2 Exposure of property to Hurricane Florence flooding**

Figure 5 shows the spatial distribution of the properties within our database overlaid with an image of the eye of Hurricane Florence when it made landfall. Our analysis indicates that the total area of properties affected by the maximum flood extent water was 70,964,700 $m^2$ (e.g., physical footprint) being 17.55 % of the total area within our database. When considering only the flood extent estimated by Sentinel-1, the total area of properties affected by the flood reduces to 3.2 %, corresponding to 12,939,432 $m^2$. In order, to quantify potential biases associated with co-registration issues or resampling procedures, we also computed the number of properties exposed to the extent of our permanent body water dataset. Our analysis shows that less than 0.2 % of properties was overlapping with the permanent body waters. Consequently, we removed these properties from our analysis.

[revised manuscript text omitted]